# Making memories last using the peripheral effect of direct current stimulation

Alison M Luckey[1], Lauren S McLeod[2], Yuefeng Huang[3], Anusha Mohan[1], Sven Vanneste[1]*

[1]Global Brain Health Institute and Institute of Neuroscience, Trinity College Dublin, Dublin, Ireland; [2]School of Medicine, Texas Tech School of Medicine, Lubbock, United States; [3]Department of Psychiatry, Icahn School of Medicine at Mount Sinai, New York, United States

**Abstract** Most memories that are formed are forgotten, while others are retained longer and are subject to memory stabilization. We show that non-invasive transcutaneous electrical stimulation of the greater occipital nerve (NITESGON) using direct current during learning elicited a long-term memory effect. However, it did not trigger an immediate effect on learning. A neurobiological model of long-term memory proposes a mechanism by which memories that are initially unstable can be strengthened through subsequent novel experiences. In a series of studies, we demonstrate NITESGON's capability to boost the retention of memories when applied shortly before, during, or shortly after the time of learning by enhancing memory consolidation via activation and communication in and between the locus coeruleus pathway and hippocampus by plausibly modulating dopaminergic input. These findings may have a significant impact for neurocognitive disorders that inhibit memory consolidation such as Alzheimer's disease.

## Editor's evaluation

This is a landmark study showing that non-invasive transcutaneous electrical stimulation of the greater occipital nerve (NITESGON) can improve long-term memory. The authors provide compelling evidence that applying NITESGON during learning causally influences memory behavior. This method is novel and the work presented herein will be valuable to a broad range of scientists and clinicians interested in manipulations to improve memory and cognition more broadly.

*For correspondence: sven.vanneste@tcd.ie

Competing interest: The authors declare that no competing interests exist.

## Introduction

Research on enhancing and preserving human memory has substantially increased in the last few decades due largely to the prevalence and inexorable condition of Alzheimer's disease (AD). Early behavioral indicators of AD include a decline in an individual's ability to retain learned information and remember events, situations, and objects, alluding to synaptic connections losing strength as AD emerges (*Reza-Zaldivar et al., 2020*). As a result, recent investigations have begun assessing the perspective clinical significance of therapeutic non-invasive brain stimulation techniques to modify neuroplasticity and upregulate neuronal excitability in different neurological conditions, including memory deficits (*Lefaucheur et al., 2017*).

There is an ongoing debate about whether non-invasive electrical stimulation of the scalp modulates the excitability of neurons directly (*Vöröslakos et al., 2018*; *Liu et al., 2018*). Interestingly, a series of experiments in rats and humans isolated the transcranial and transcutaneous mechanisms of

non-invasive electrical stimulation and showed that the reported effects are mainly caused by transcutaneous stimulation of peripheral nerves (*Vöröslakos et al., 2018*; *Asamoah et al., 2019*). Similarly, it was demonstrated that nerve stimulation paired with an auditory or motor task could induce targeted plasticity in animals (*Engineer et al., 2011*; *Porter et al., 2012*). In addition, our recent work suggests that non-invasive transcutaneous electrical stimulation of the greater occipital nerve (NITESGON) using direct current utilizes a pathway that arises from the C2 spinal nerve to establish communication gateways from the periphery to the brain via afferent fibers that project to the brainstem and synapse onto the nucleus tractus solitarius (NTS); various rodent studies have used anterograde neuronal tracers to demonstrate these connections (*Adair et al., 2020*; *Caous et al., 2001*; *de Sousa Buck et al., 2001*; *Menétrey and Basbaum, 1987*). From the NTS, the information is then integrated among networks within the complex reticular formation and relayed across the brainstem through major cortical and subcortical regions (*Kawai, 2018*). Furthermore, our work demonstrated that NITESGON during learning induces improvements in memory recall in younger (18–25 years) and older (>55 years) adults up to 28 days after learning (*Luckey et al., 2020*; *Vanneste et al., 2020*). Intriguingly, NITESGON yielded a long-term memory effect but did not trigger an immediate effect on learning, suggesting that the effect is generated during the consolidation of memories (*Vanneste et al., 2020*; *Luckey et al., 2020*) as opposed to during the learning or encoding of new memories.

Most episodic-like memories that are formed are forgotten, while others are retained for longer periods of time and are subject to memory stabilization (*Squire, 1992*; *Squire et al., 1992*; *Tonegawa et al., 2018*). This is referred to as synaptic consolidation, a process that stabilizes new information into memory over a timespan of minutes to hours (*Squire et al., 2015*). To illustrate the neurobiological account of synaptic consolidation, Frey and Morris introduced a plasticity model known as the synaptic tag-and-capture hypothesis that has proposed a cellular mechanism explaining how new memories that are initially weak and unstable are tagged to be captured by late-phase long-term potentiation (LTP) to become stable (*Morris and Frey, 1997*; *Frey and Morris, 1997*). This synaptic tag-and-capture mechanism has since been translated into a learning and memory paradigm referred to as behavioral tagging (*Moncada et al., 2015*; *Viola et al., 2014*). Behavioral tagging proposes weak training that typically generates short-term memory can utilize the tag-and-capture process to consolidate into a stabilized long-term memory when a weak event is preceded or followed by a strong or novel event within a limited time window (*Moncada et al., 2015*; *Viola et al., 2014*; *Dunsmoor et al., 2022*). The neural mechanism that controls this novelty response is the locus coeruleus–noradrenaline (LC–NA) pathway (*Vankov et al., 1995*; *Takeuchi et al., 2016*). Animal research further indicates that direct electrical stimulation of the LC modulates hippocampal synaptic consolidation (*Lemon and Manahan-Vaughan, 2012*; *McGaugh, 2004*; *Cahill and McGaugh, 1998*). We hypothesize that NITESGON modulates projections to the hippocampus via the LC–NA system and induces memory stabilization by modulating synaptic consolidation in the hippocampus via the mechanism of behavioral tagging.

The present study tests the above hypothesis questioning if NITESGON induces a long-term memory effect by strengthening memories via behavioral tagging across eight experiments. The first set of experiments aims to confirm the behavioral tagging hypothesis as the potential mechanism inducing memory consolidation via NITESGON, whereby the second set of experiments examines the underlying brain network involved in synaptic consolidation and investigates the underlying neural mechanism that is associated with behavioral tagging induced by NITESGON.

## Results

### Experiment 1. NITESGON during or immediately after training

The idea behind behavioral tagging suggests that weak memories that are regularly unstable and likely to be forgotten will solidify following a novel experience (*Moncada et al., 2015*; *Viola et al., 2014*; *Dunsmoor et al., 2022*). That is, consolidation is facilitated by applying a strong stimulus alongside a weak stimulus within a critical time window. Recent research revealed a direct link between the LC and behavioral tagging, attributable to the pivotal role the LC plays during the presentation of a salient or arousing event (i.e., strong stimulus) (*Sara, 2009*; *Poe et al., 2020*), as well as being at the helm of regulating the synthesis of new proteins required for memory consolidation in the hippocampus (*Moncada, 2017*). Furthermore, studies have shown modulation of memory consolidation

with increases in stress and arousal that are mediated via the LC pathway (*McGaugh, 2004*; *Cahill and McGaugh, 1998*). Moreover, animal research has indicated that direct electrical stimulation of the LC modulates hippocampal synaptic transmission fundamental for memory consolidation (*Lemon and Manahan-Vaughan, 2012*).

Seeing that NITESGON activates the LC pathway, which plays an important role in memory consolidation, we hypothesize that participants will be able to establish long-term memories upon modulating the LC both during learning, as shown before (*Luckey et al., 2020*; *Vanneste et al., 2020*), as well as immediately after learning. This will directly test if NITESGON plays a more central role during encoding or the consolidation phase of memory. To test this hypothesis, participants learned a word-association task and were tested 7 days later on how many word associations they were able to recall correctly. Active or sham NITESGON was applied via electrodes placed over the left and right C2 nerve dermatome at a constant current of 1.5 mA either during or immediately after learning the word-association task on visit 1.

To further explore the effect of NITESGON, resting-state EEG (rsEEG) and salivary α-amylase (sAA) were collected immediately before and after NITESGON on visit 1. Previous research has revealed an increase in sAA, a putative marker of endogenous NA activity, immediately following NITESGON (*Luckey et al., 2020*; *Vanneste et al., 2020*). Furthermore, previous investigations have demonstrated that LC discharge enhances the synchronization of gamma activity in the hippocampus in rats (*Hajós et al., 2003*) and have identified gamma oscillations' critical role in long-term memory formation and the potential to predict subsequent recall (*Osipova et al., 2006*; *Sederberg et al., 2003*). Based on these findings, we hypothesized that NITESGON would induce an increase in sAA and gamma activity in the medial temporal lobe that will correlate with successful recall during the second visit 7 days after learning the task.

On visit 1, no difference was observed regarding the number of word associations learned between the three condition groups (i.e., sham NITESGON during learning and after learning, active NITESGON during learning and sham NITESGON after learning, or sham NITESGON during learning and active NITESGON after learning) ($F = 0.24$, $p = 0.79$; see *Figure 1a*), thus indicating that NITESGON had no effect on learning the word-association task. Results revealed a significant difference in memory recall 7 days after NITESGON was applied either during learning ($46.09 \pm 15.06\%$, $p = 0.012$) or after learning the task ($47.65 \pm 13.27\%$, $p = 0.005$) relative to the sham condition employed during both learning and immediately after learning the task ($33.38 \pm 12.57\%$) ($F = 5.24$, $p = 0.009$; see *Figure 1b*). However, no difference was attained on recall 7 days later between the conditions of NITESGON applied during learning the task or immediately after learning the task ($p = 0.75$). A significant increase in sAA ($F = 7.69$, $p = 0.010$; see *Figure 1c*) was revealed during learning (before: $88.79 \pm 50.48$ vs. after: $149.82.6 \pm 82.67$; $p < 0.001$) and after learning in comparison to the sham group (before: $100.28 \pm 41.95$ vs. after: $114.30 \pm 41.02$; $p = 0.14$). Memory recall 7 days later correlated with the difference in sAA levels on visit 1 (pre vs. post) ($r = 0.59$, $p < 0.001$; *Figure 1d*; $r_s = 0.62$, $p < 0.001$; *Figure 1—figure supplement 1*). Looking at the individual correlations for each group separately, we found a significant correlation for the active groups (i.e., NITESGON during learning $r = 0.47$, $p = 0.09$ and after learning [$r = 0.78$, $p < 0.001$]) between memory recall 7 days later and sAA levels on visit 1 (pre vs. post). No significant correlation was obtained for the sham group ($r = 0.27$, $p = 0.31$). Memory recollection 7 days after stimulation was associated with increased gamma power in the medial temporal cortex as well as the precuneus and dorsal lateral prefrontal cortex immediately after stimulation ($r = 0.42$, $p = 0.011$; see *Figure 1e*).

## Experiment 2. NITESGON during second task – retroactive strengthening of memories

Experiment 1 suggests that NITESGON generates an effect during the consolidation phase as opposed to the learning-encoding phase due to no effect of NITESGON being exhibited during learning between the different groups, but both stimulating during and after learning the task induced a long-term memory effect. Bearing in mind the definition of behavioral tagging that indicates that the pairing of a strong stimulus and a weak stimulus within a critical time window can induce memory stabilization of the weak stimulus, NITESGON can be seen as the mechanism that induces a similar action as a strong stimulus, and through the mechanism of behavioral tagging strengthen the weak stimulus (i.e., the word-association task).

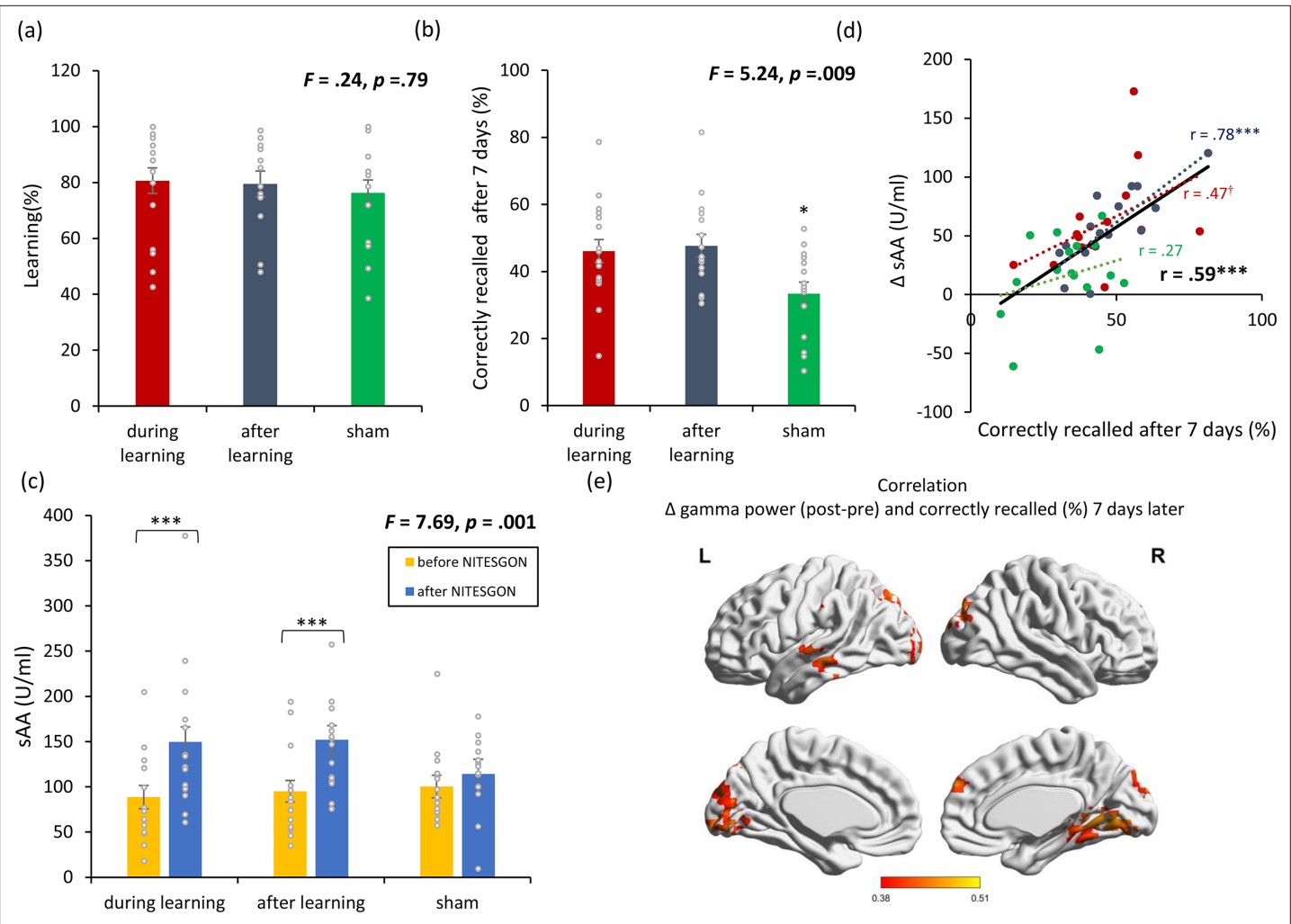

**Figure 1.** Non-invasive transcutaneous electrical stimulation of the greater occipital nerve (NITESGON) immediately after training can enhance memory with *Figure 1—figure supplement 1*. (**a**) No difference was observed in the cumulative learning rate between active and sham NITESGON during or immediately after the study phase of the word-association memory task. (**b**) NITESGON during or immediately after the word-association memory task can improve memory recall 7 days after the study phase for the active relative to the sham group. (**c**) After NITESGON salivary α-amylase (sAA) levels increase for both active groups, but not for sham NITESGON. (**d**) Memory recall 7 days later correlates with the difference in sAA levels during the first visit (pre vs. post study phase). (**e**) Improved memory recall 7 days after stimulation is associated with increased activity in the medial temporal lobe as well as anterior and posterior cingulate cortex immediately after NITESGON for the gamma frequency band. Error bars, standard error of the mean (s.e.m.). Asterisks represent significant differences (*p < 0.05; **p < 0.01; *** p < 0.001), ΔsAA levels are the subtraction of sAA levels before NITESGON from sAA levels immediately after NITESGON.

The online version of this article includes the following figure supplement(s) for figure 1:

**Figure supplement 1.** To avoid the potential outliers would drive the Pearson correlation between a memory recall 7 days after learning the task and the difference in salivary α-amylase (sAA) levels (difference between before and after stimulation on day 1), a Spearman rank correlation was calculated to cross validate our findings.

Prior research on behavioral tagging has shown that items paired with an electric shock (i.e., Pavlovian fear conditioning task) had a retroactive memory effect on items learned before the fear conditioning task (*Dunsmoor et al., 2015*). This provided evidence for a generalized retroactive memory enhancement, whereby information can be retroactively credited as relevant, and therefore remembered (*Dunsmoor et al., 2015*). Interestingly, LC activation occurs in close relation to the intensity of the Pavlovian behavior (*Bouret and Richmond, 2009*). Hence, to explore the effect of the LC on behavioral tagging, we verified if NITESGON applied during a second task would result in a significant retroactive memory effect on the first task as predicted by behavioral tagging.

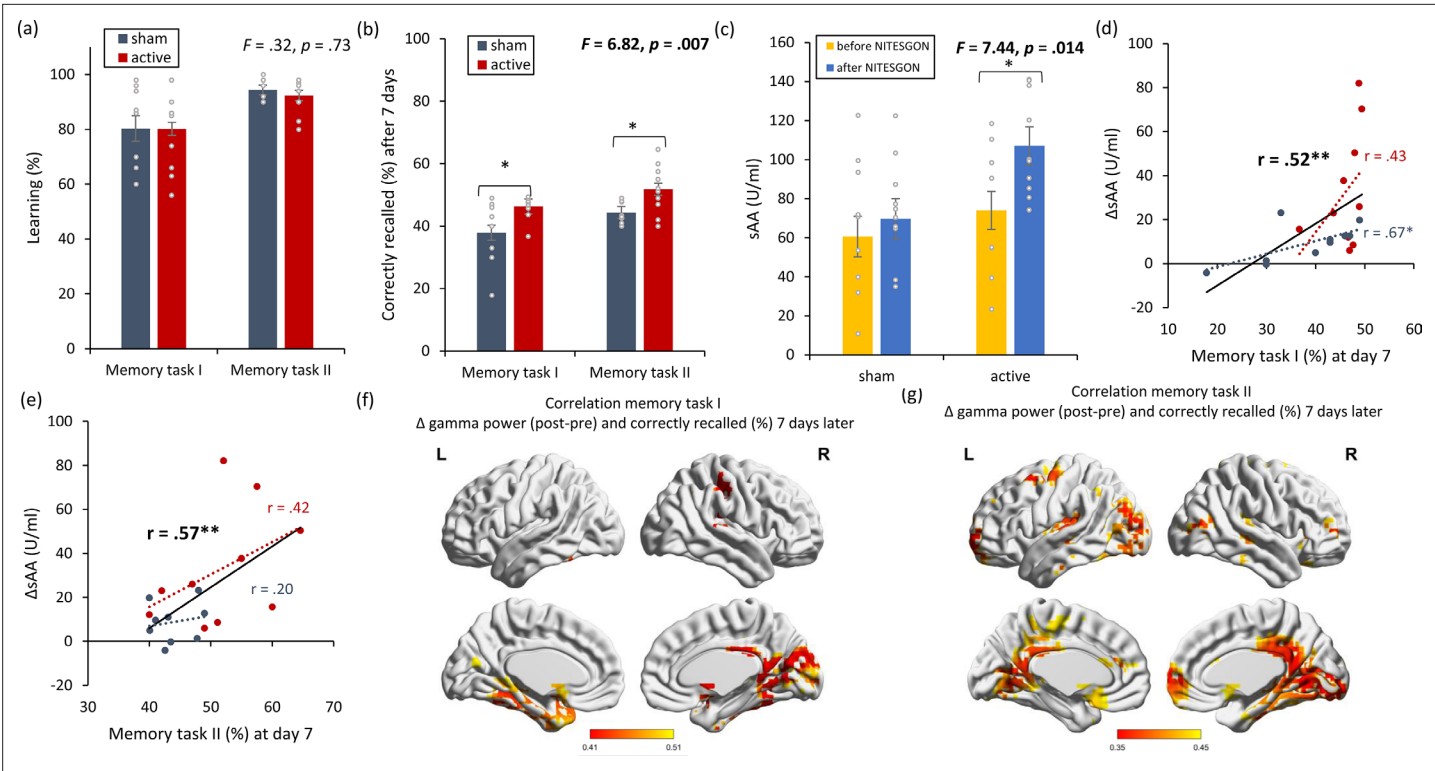

**Figure 2.** Non-invasive transcutaneous electrical stimulation of the greater occipital nerve (NITESGON) has a retroactive memory effect – NITESGON during the second task memory with **Figure 2—figure supplement 1**. (**a**) No difference was observed in the cumulative learning rate between active and sham NITESGON after the study phase for the first task (i.e., word-association task) or second task (i.e., object-location task). (**b**) NITESGON can improve memory recall 7 days after the study phase for the active relative to the sham group for both the first and second tasks. (**c**) After NITESGON salivary α-amylase (sAA) levels increase for active group, but not for sham NITESGON. (**d, e**) Memory recall 7 days later correlates with the difference in sAA levels during the first visit (pre vs. post study phase) for the first and second tasks. (**f, g**) Improved memory recall 7 days after stimulation is associated with increased activity in the medial temporal lobe immediately after NITESGON for the gamma frequency band. Error bars, standard error of the mean (s.e.m.). Asterisks represent significant differences (*p < 0.05; **p < 0.01). ΔsAA levels are the subtraction of sAA levels before NITESGON from sAA levels immediately after NITESGON.

The online version of this article includes the following figure supplement(s) for figure 2:

**Figure supplement 1.** To avoid the potential outliers would drive the Pearson correlation between a memory recall 7 days after learning the task and the difference in salivary α-amylase (sAA) levels (difference between before and after stimulation on day 1), a Spearman rank correlation was calculated to cross validate our findings.

To test the hypothesis, Experiment 2 had participants take part in a word-association task followed by a spatial-navigation object-location task while receiving active or sham NITESGON during the second task. These two types of tasks were selected because they would not interfere with one another, seeing that both require different episodic information. rsEEG data and sAA were collected immediately before and after the two tasks on visit 1. Two memory tests were taken 7 days after learning the word-association and spatial-navigation tasks.

On visit 1, no difference in learning ($F = 0.32$, p = 0.73; see **Figure 2a**) was observed for both the first ($F = 0.09$, p = 0.98) and second ($F = 0.64$, p = 0.43) tasks between the active and sham NITESGON groups. On visit 2, 7 days after initial learning, a significant effect was obtained for recall ($F = 0.6.82$, p = 0.007; see **Figure 2b**) for both the first ($F = 6.28$, p = 0.022) and second tasks ($F = 7.51$, p = 0.013), revealing an increase in word recall (46.26 ± 3.76% vs. 37.88 ± 9.88%), as well as object-location recall (51.82 ± 7.75% vs. 44.39 ± 3.68%) for the active group in comparison to the sham group. Furthermore, a significant increase in sAA ($F = 7.44$, p = 0.014; see **Figure 2c**) was revealed in the active group (before: 74.01 ± 26.58 vs. after: 107.01 ± 25.98; p < 0.001) in comparison to the sham group (before: 60.61 ± 37.93 vs. after: 69.61 ± 35.07; p = 0.18). This increase in sAA correlated with how many items they recalled 7 days after the learning phase for both the word-association task (r = 0.52, p = 0.019; see **Figure 2d**; $r_s$ = 0.62, p = 0.002; **Figure 2—figure supplement 1**) and the object-location task (r =

0.57, p = 0.008; see *Figure 2e*; $r_s$ = 0.49, p = 0.03 *Figure 2—figure supplement 1*). Looking at each group separately, we found no significant correlation for the active groups (first task: *r* = 0.43, p = 0.21; second task: *r* = 0.42, p = 0.22) between memory recall 7 days later and sAA levels on visit 1 (pre vs. post). For the sham group a significant effect was obtained between memory recall 7 days later and sAA levels on visit 1 (pre vs. post) for the first task (*r* = 0.67, p = 0.036). No significant correlation was obtained for the sham group for the second task (*r* = 0.20, p = 0.58). Memory recollection 7 days after stimulation was associated with increased gamma power in the medial temporal cortex immediately after stimulation for both the first (*r* = 0.41, p = 0.009; see *Figure 2f*) and second memory tasks (*r* = 0.35, p = 0.018; see *Figure 2g*).

## Experiment 3. NITESGON during first task – proactive strengthening of memories

Experiment 2 revealed a retroactive memory effect 7 days after initial learning for the active NITESGON group compared to the sham NITESGON group, fitting well with the behavioral tagging hypothesis. In addition to a retroactive memory effect, previous research on behavioral tagging also revealed that items paired with an electric shock had a proactive memory effect, whereby items learned after the fear conditioning task were remembered (*Dunsmoor et al., 2015*). Here, we conducted the exact

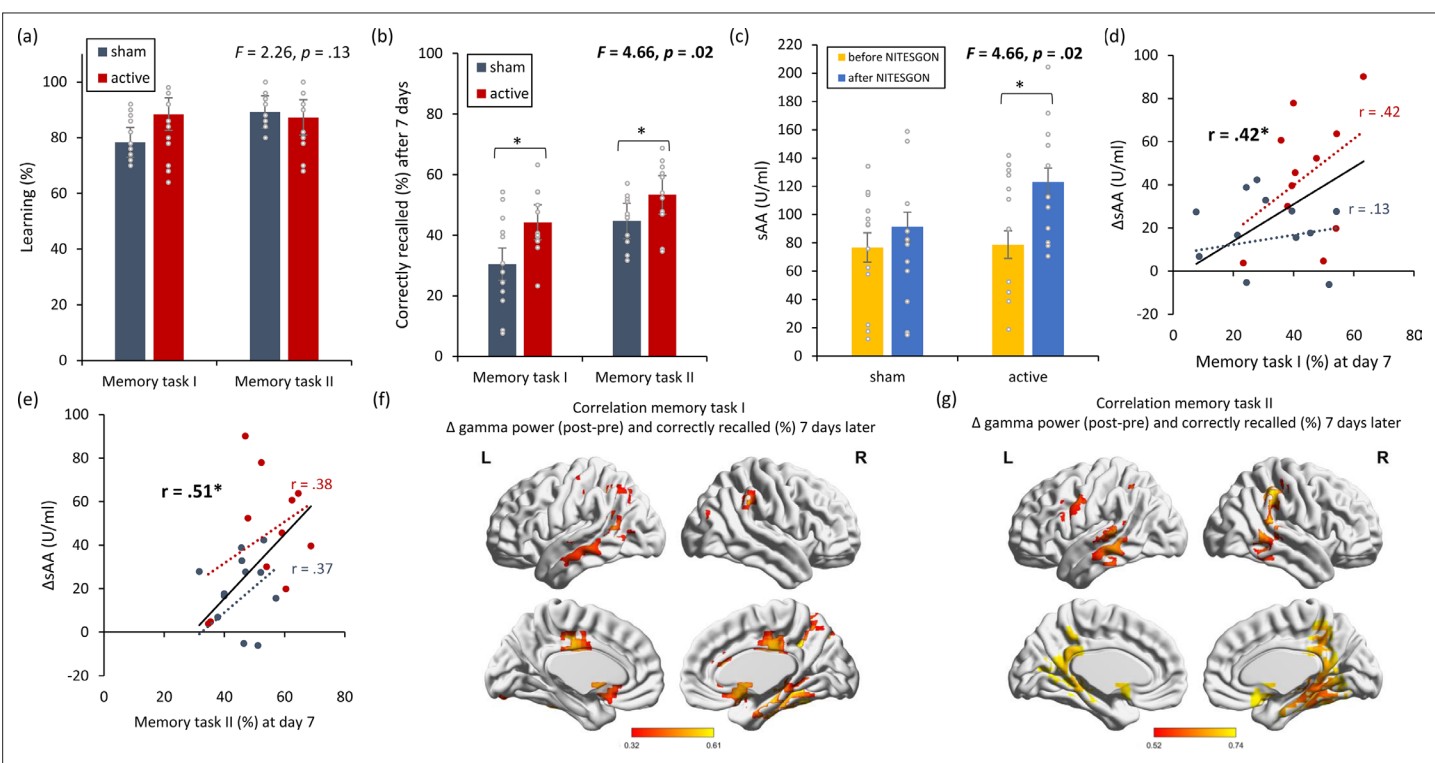

**Figure 3.** Non-invasive transcutaneous electrical stimulation of the greater occipital nerve (NITESGON) has a proactive memory effect – NITESGON during the first task memory with *Figure 3—figure supplement 1*. (**a**) No difference was observed in the cumulative learning rate between active and sham NITESGON after the study phase for the first task (i.e., word-association task) or second task (i.e., object-location task). (**b**) NITESGON can improve memory recall 7 days after the study phase for the active relative to the sham group for both the first and second tasks. (**c**) After NITESGON salivary α-amylase (sAA) levels increase for active group, but not for sham NITESGON. (**d, e**) Memory recall 7 days later correlates with the difference in sAA levels during the first visit (pre vs. post study phase) for the first and second tasks. (**f, g**) Improved memory recall 7 days after stimulation is associated with increased activity in the medial temporal lobe immediately after NITESGON for the gamma frequency band. Error bars, standard error of the mean (s.e.m.). Asterisks represent significant differences (*p < 0.05; **p < 0.01), ΔsAA levels are the subtraction of sAA levels before NITESGON from sAA levels immediately after NITESGON.

The online version of this article includes the following figure supplement(s) for figure 3:

**Figure supplement 1.** To avoid the potential outliers would drive the Pearson correlation between a memory recall 7 days after learning the task and the difference in salivary α-amylase (sAA) levels (difference between before and after stimulation on day 1), a Spearman rank correlation was calculated to cross validate our findings.

same experiment as in Experiment 2 but applied NITESGON during the first task and not during the second task to test the hypothesis that NITESGON can induce a proactive memory effect on the second task although we stimulate during the first task. This would further support the hypothesis that NITESGON induces a long-term memory effect via the mechanism of behavioral tagging through activation of the LC pathway.

On visit 1, no significant difference ($F$ = 2.26, p = 0.13; see *Figure 3a*) was found between the active and sham groups in regard to how many words or objects participants learned for both the first task (i.e., word-association task) ($F$ = 1.60, p = 0.22) and the second task (i.e., object-location task) ($F$ = 3.30, p = 0.08). During the second visit, 7 days after learning the tasks, participants that received active NITESGON ($F$ = 4.66, p = 0.021; see *Figure 3b*) recalled more words for the first task (i.e., word-association task) ($F$ = 6.32, p = 0.020) and the second task (i.e., object-location task) ($F$ = 4.87, p = 0.038) than those who received sham NITESGON, indicating that the active NITESGON group (44.27 ± 10.97) showed significant improvement in comparison to the sham NITESGON group (30.46 ± 15.13) for the word-association task. For the object-location task, the active NITESGON group (53.33 ± 11.29) demonstrated a significant increase in the number of correctly recalled objects-locations than the sham NITESGON group (44.17 ± 7.75). Our data revealed that there was a significant interaction effect for sAA ($F$ = 4.66, p = 0.021; see *Figure 3c*), denoted by the active group's increase in sAA (123.10 ± 43.63) in comparison to the sham group (91.38 ± 44.67) ($F$ = 4.53, p = 0.039) immediately after learning. No significant difference ($F$ = 0.012, p = 0.91) was obtained in sAA for the active group (78.72 ± 47.67) in comparison to the sham group (76.77 ± 39.87) before learning the association tasks. This increase in sAA seen in the active group correlated with how many items they recalled 7 days after the learning phase for both the word-association task ($r$ = 0.42, p = 0.039; see *Figure 3d*; $r_s$ = 0.34, p = 0.10 *Figure 3—figure supplement 1*; a correlation analysis for each group separately) and the object-location task ($r$ = 0.51, p = 0.012; see *Figure 3e*; $r_s$ = 0.54, p = 0.007 *Figure 3—figure supplement 1*). Looking at each group separately, we found no significant correlation for both the active (first task: $r$ = 0.42, p = 0.20; second task: $r$ = 0.38, p = 0.25) and sham groups (first task: $r$ = 0.13, p = 0.21; second task: $r$ = 0.37, p = 0.22) between memory recall 7 days later and sAA levels on visit 1 (pre vs. post). Memory recollection 7 days after stimulation was associated with increased gamma power in the medial temporal cortex immediately after stimulation for both the first ($r$ = 0.32, p = 0.037; see *Figure 3f*) and second memory tasks ($r$ = 0.52, p = 0.012; see *Figure 3g*).

## Experiment 4. NITESGON during first task – reduced interference effect

Experiments 2 and 3 revealed both retroactive and proactive memory effects 7 days after initial learning of the two tasks. To further explore if NITESGON is linked to behavioral tagging and evaluate if interference impacts NITESGON as the strong stimulus, Experiment 4 removed the object-location task used in Experiments 2 and 3 and replaced it with a Japanese–English verbal associative learning task similar to the Swahili–English verbal associative task. Considering that memory formation and persistence are susceptible to interference occurring pre- and post-encoding (*Crossley et al., 2019*; *McGaugh, 1966*; *Zeithamova and Preston, 2017*) and are heavily influenced by commonality among the learned and intervening stimuli (*Varma et al., 2017*) it is believed that conducting two consecutive, like-minded word-association (i.e., Swahili–English and Japanese–English) tasks will result in one's consolidation process interfering with that of the other (*Robertson, 2012*). Furthermore, research on the synaptic tag-and-capture hypothesis suggests that memory interference is the result of synaptic competition, a proposed 'fight for proteins' that arises between tagged synapses among limited proteins that leads to one memory converting to long-term memory at the expense of the other (*Okuda et al., 2021*). Considering how our previous experiments suggest the effect obtained by NITESGON improves the consolidation of information via behavioral tagging, it is possible that NITESGON on the first task might help reduce the overall interference effect on the second task.

To test the hypothesis, participants participated in two separate word-association tasks (i.e., the Swahili–English and Japanese–English; the order of tasks was randomized across participants) while receiving either active or sham NITESGON during the first task. rsEEG data and sAA were collected immediately before and immediately after the two tasks on visit 1. Two memory tests were taken 7 days after learning both word-association tasks.

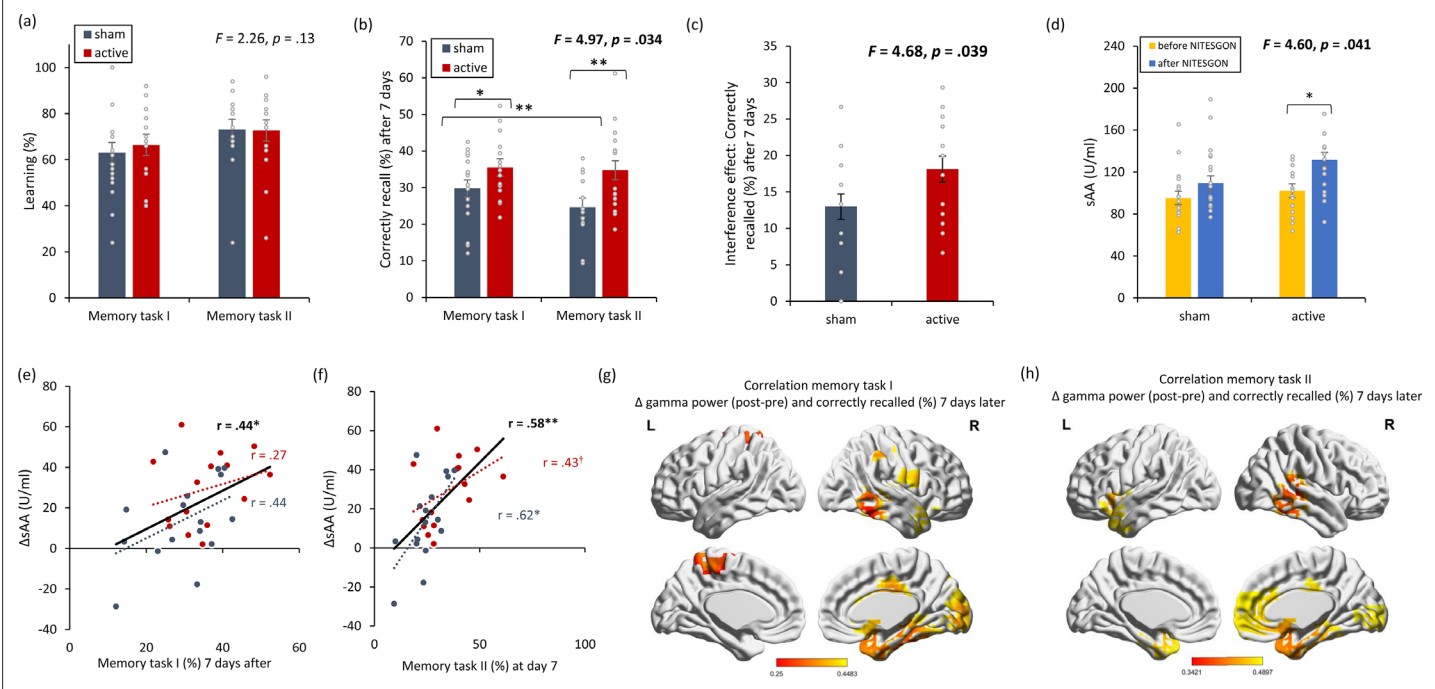

**Figure 4.** Non-invasive transcutaneous electrical stimulation of the greater occipital nerve (NITESGON) reduces the interference effect memory with *Figure 4—figure supplement 1*. (**a**) No difference was observed in the cumulative learning rate between active and sham NITESGON after the study phase for the first task (i.e., word-association task) or second task (i.e., word-association task). (**b**) NITESGON can improve memory recall 7 days after the study phase revealing improve in memory for the active relative to the sham group for both the first and second tasks. (**c**) The interference effect is less present for the active relative to the sham group. (**d**) After NITESGON salivary α-amylase (sAA) levels increase for the active group, but not for sham NITESGON. (**e, f**) Memory recall 7 days later correlates with the difference in sAA levels during the first visit (pre vs. post study phase) for the first and second tasks. (**g, h**) Improved memory recall 7 days after stimulation is associated with increased activity in the medial temporal lobe immediately after NITESGON for the gamma frequency band. Error bars, standard error of the mean (s.e.m.). Asterisks represent significant differences (*p < 0.05; **p < 0.01), ΔsAA levels are the subtraction of salivary α-amylase levels before NITESGON from salivary α-amylase levels immediately after NITESGON.

The online version of this article includes the following figure supplement(s) for figure 4:

**Figure supplement 1.** To avoid the potential outliers would drive the Pearson correlation between a memory recall 7 days after learning the task and the difference in salivary α-amylase (sAA) levels (difference between before and after stimulation on day 1), a Spearman rank correlation was calculated to cross validate our findings.

On visit 1, no significant difference ($F = 0.84$, $p = 0.37$; see *Figure 4a*) was detected for learning during the first ($F = 0.27$, $p = 0.61$) and second tasks ($F = 0.01$, $p = 0.94$) between the active and sham NITESGON groups. Seven days later, during the recall phase, we found a significant interaction effect for recall ($F = 4.97$, $p = 0.034$; see *Figure 4b*). For both the first ($F = 3.67$, $p = 0.048$) and the second tasks ($F = 7.89$, $p = 0.009$), a significant increase in number of words correctly recalled was observed in the active (first task: 35.51 ± 8.68; second task: 34.76 ± 11.74) compared to the sham group (first task: 29.80 ± 9.72; second task: 24.64 ± 8.10).

Upon assessment for a potential interference effect, the active group displayed a significant difference in how many words participants were able to recall between the first and the second tasks (18.31 ± 7.03%) ($F = 4.68$, $p = 0.039$) in comparison to the sham (13.00 ± 6.65%), indicating that the interference effect plays less of a role in the active group.

Our data revealed that there was a significant interaction effect for sAA ($F = 4.60$, $p = 0.041$; see *Figure 4d*). An increase in sAA was observed for the active group (123.10 ± 43.63) in comparison to the sham group (91.38 ± 44.67) ($F = 4.53$, $p = 0.039$) immediately after the learning. However, no significant difference ($F = 0.012$, $p = 0.91$) was obtained in sAA for the active group (78.72 ± 47.67) in comparison to the sham group (76.77 ± 39.87) before learning the word-association tasks. This increase in sAA correlates with how many words they recalled 7 days after the learning phase for both the first word-association task ($r = 0.44$, $p = 0.014$; see *Figure 4e*; $r_s = 0.35$, $p = 0.049$ *Figure 4—figure supplement 1*) and the second word-association task ($r = 0.58$, $p = 0.001$; see *Figure 4f*; $r_s$

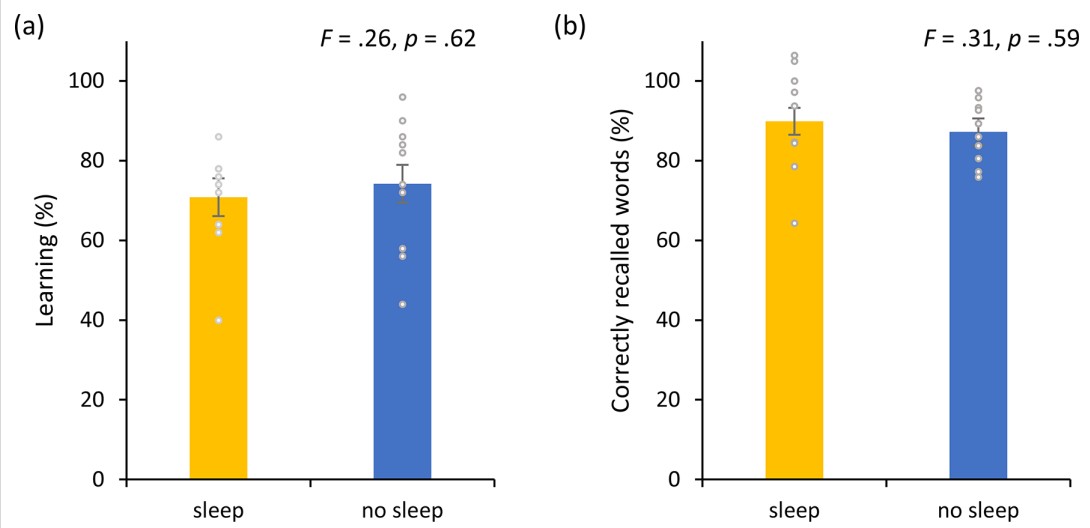

**Figure 5.** Non-invasive transcutaneous electrical stimulation of the greater occipital nerve (NITESGON) and sleep. (**a**) No difference was observed in the cumulative learning rate between participants who had slept versus those who had not slept after NITESGON applied during the study phase. (**b**) Sleep has no effect on memory recall 12 hr after the study phase. Error bars, standard error of the mean (s.e.m.). Asterisks represent significant differences (*p < 0.05; **p < 0.01).

= 0.53, p = 0.002 *Figure 4—figure supplement 1*). Looking at each group separately, we found no significant correlation for either the active ($r = 0.27$, $p = 0.34$) or sham group ($r = 0.42$, $p = 0.11$) between memory recall 7 days later and sAA levels on visit 1 (pre vs. post) for the first task. For the second task, we found a significant correlation for both the active ($r = 0.43$, $p = 0.098$) and sham groups ($r = 0.62$, $p = 0.010$) was obtained between memory recall 7 days later and sAA levels on visit 1 (pre vs. post) for the first task.

Memory recollection 7 days after stimulation was also associated with increased gamma power in the medial temporal cortex immediately after stimulation for both the first ($r = 0.25$, $p = 0.042$; see *Figure 4g*) and second memory tasks ($r = 0.34$, $p = 0.032$; see *Figure 4h*).

## Experiment 5. The effect of NITESGON is not sleep dependent

Our behavioral experiments suggest that NITESGON targeting the LC is involved in synaptic consolidation via the behavioral tagging mechanism. It is assumed that synaptic consolidation occurs over a timespan of minutes to hours after encoding the information, thus this effect is time dependent (*Park, 2005*). Furthermore, prior research has revealed that retroactive memory enhancement (i.e., evidence for behavioral tagging) emerges within 6 hr and is not dependent on sleep (*Dunsmoor et al., 2015*). Based on these previous findings and the assumption that NITESGON modulates synaptic consolidation via the mechanism of behavioral tagging, we hypothesize that sleep would not mediate the memory effect induced by NITESGON.

Experiment 5 compared two groups of participants undertaking a word-association task paired with active NITESGON. One group of participants slept between the word-association task at 8 p.m. and the test phase the next day at 8 a.m., whereas the other group did not sleep between the learning phase at 8 a.m. and the test phase that took place at 8 p.m. that same day. A comparison between the two groups revealed no significant difference in the number of words learned during the learning phase ($F = 0.26$, $p = 0.62$; see *Figure 5a*) as well as no significant difference between the two groups when tested 12 hr later ($F = 0.31$, $p = 0.59$; see *Figure 5b*). Participants who slept in-between the learning and test phase correctly recalled 89.99 ± 13.09% of word pairs and participants who did not sleep in-between the learning and test phase correctly recalled 87.23 ± 7.41% of word pairs.

## Experiment 6. LC – hippocampus activity and connectivity

In addition to our behavioral experiments confirming the hypothesis that NITESGON targeting the LC is involved in memory consolidation via the behavioral tagging mechanism, Experiment 5 revealed that sleep does not mediate the effect generated by NITESGON. Here, in the second set of experiments,

we explored the brain network modulated by NITESGON and investigated the potential underlying neural mechanism.

The hippocampus is the key brain area that has been associated with synaptic memory consolidation. This area receives neuromodulatory input from multiple brain regions that regulate synaptic plasticity, such as the LC and the ventral tegmental area (VTA). Both brain areas enhance retention of everyday memories in the hippocampus (*Takeuchi et al., 2016*). Moreover, animal research identified the VTA and the LC as regulators of hippocampal-dependent long-term memory formation due to their role in regulating the synthesis of new proteins required during the behavioral tagging process, therefore allowing for the consolidation of lasting memories (*Moncada, 2017*). However, recent studies have shown that the VTA projections to the hippocampus are scarce, while the LC projections are abundant (*Takeuchi et al., 2016*; *Kempadoo et al., 2016*; *McNamara et al., 2014*). Therefore, the VTA may only play a limited role in late-phase LTP (*Takeuchi et al., 2016*; *Kempadoo et al., 2016*; *McNamara et al., 2014*), whereas the LC is conceivably the primary source of synaptic modulation responsible for tuning cells in the hippocampus (*Kempadoo et al., 2016*). Additionally, several previous observations have shown electrical and pharmacological stimulation of the LC modulated hippocampal synaptic transmission (*Lemon and Manahan-Vaughan, 2012*; *Devoto and Flore, 2006*), whilst modulation of the VTA did not significantly mediate synaptic transmission but rather suggest a greater role in salience and motivational drive underlying emotion-based learning (*Kempadoo et al., 2016*; *Devoto and Flore, 2006*).

We conducted a resting-state functional connectivity magnetic resonance imaging study to verify the relationship between changes in the LC and hippocampus as well as the VTA and hippocampus. We hypothesized that participants who received active NITESGON would show increased activity in the LC and hippocampus, but not in the VTA, and increased functional connectivity between the LC and hippocampus, but not between the VTA and hippocampus. We scanned in three consecutive blocks: immediately before, during, and immediately after stimulation. NITESGON was applied at a constant current of 1.5 mA for 20 min via electrodes placed over the left and right C2 nerve dermatome.

The regional amplitude of low-frequency fluctuations (rALFF) was inspected to verify if NITESGON evoked activity changes in the LC, VTA, and hippocampus. Our findings showed a significant effect for the LC ($F = 4.34$, $p = 0.023$), VTA ($F = 3.42$, $p = 0.047$) and hippocampus ($F = 3.52$, $p = 0.044$) when comparing the active and control groups (see *Figure 6a–c*). For both the LC and hippocampus, a significant increase was obtained during (LC: $13.18 \pm 4.18$ vs. $8.77 \pm 2.88$; $F = 11.30$, $p = 0.002$; hippocampus: $6.30 \pm 3.55$ vs. $4.50 \pm 2.88$; $p = 0.045$) and after (LC: $13.78 \pm 6.21$ vs. $6.71 \pm 2.79$; $p < 0.001$; hippocampus: $7.17 \pm 4.61$ vs. $4.33 \pm 1.44$; $p = 0.031$) stimulation for the active group in comparison to the sham group. Before stimulation, no significant difference was obtained between the active and sham groups (LC: $9.40 \pm 4.50$ vs. $8.98 \pm 1.92$; $p = 0.76$; hippocampus: $5.26 \pm 2.20$ vs. $5.50 \pm 1.64$; $p = 0.75$). For the VTA, a significant increase was obtained during ($20.01 \pm 5.90$ vs. $14.12 \pm 5.34$; $p = 0.008$) stimulation for the active group in comparison to the sham group. Before ($14.96 \pm 4.50$ vs. $14.96 \pm 1.92$; $p = 0.99$) or after ($15.33 \pm 4.98$ vs. $15.46 \pm 13.86$; $p = 0.97$) stimulation, no significant difference was obtained between the active and sham groups. To further confirm our data, we replicated our analysis by not including a smoothing kernel, showing similar results for the LC, VTA, and hippocampus. Furthermore, we included two control areas, the left inferior parietal cortex, where we do not expect to see any changes (see *Figure 6—figure supplement 1*).

Furthermore, a regions of interest (ROI)-to-ROI analysis demonstrated an effect between the right hippocampus and LC ($F = 3.67$, $p = 0.039$), but not between the right hippocampus and VTA ($F = 0.27$, $p = 0.76$) (see *Figure 1d,e*). Additionally, an increase in LC connectivity strength with the right hippocampus was seen for the active group relative to the sham group during ($0.052 \pm 0.03$ vs. $0.018 \pm 0.06$; $F = 4.34$, $p = 0.047$) and after stimulation ($0.06 \pm 0.05$ vs. $-0.011 \pm 0.05$; $F = 15.25$, $p = 0.001$). However, no significant effect was obtained between the LC and right hippocampus for the active group relative to the sham group before stimulation ($0.008 \pm 0.08$ vs. $0.015 \pm 0.03$; $F = 0.09$, $p = 0.76$).

## Experiment 7. Potential relationship between NITESGON-LC and dopamine

The previous experiment revealed activity changes in both the LC and hippocampus as well as increased connectivity between the LC and hippocampus both during and after NITESGON. Conversely, the

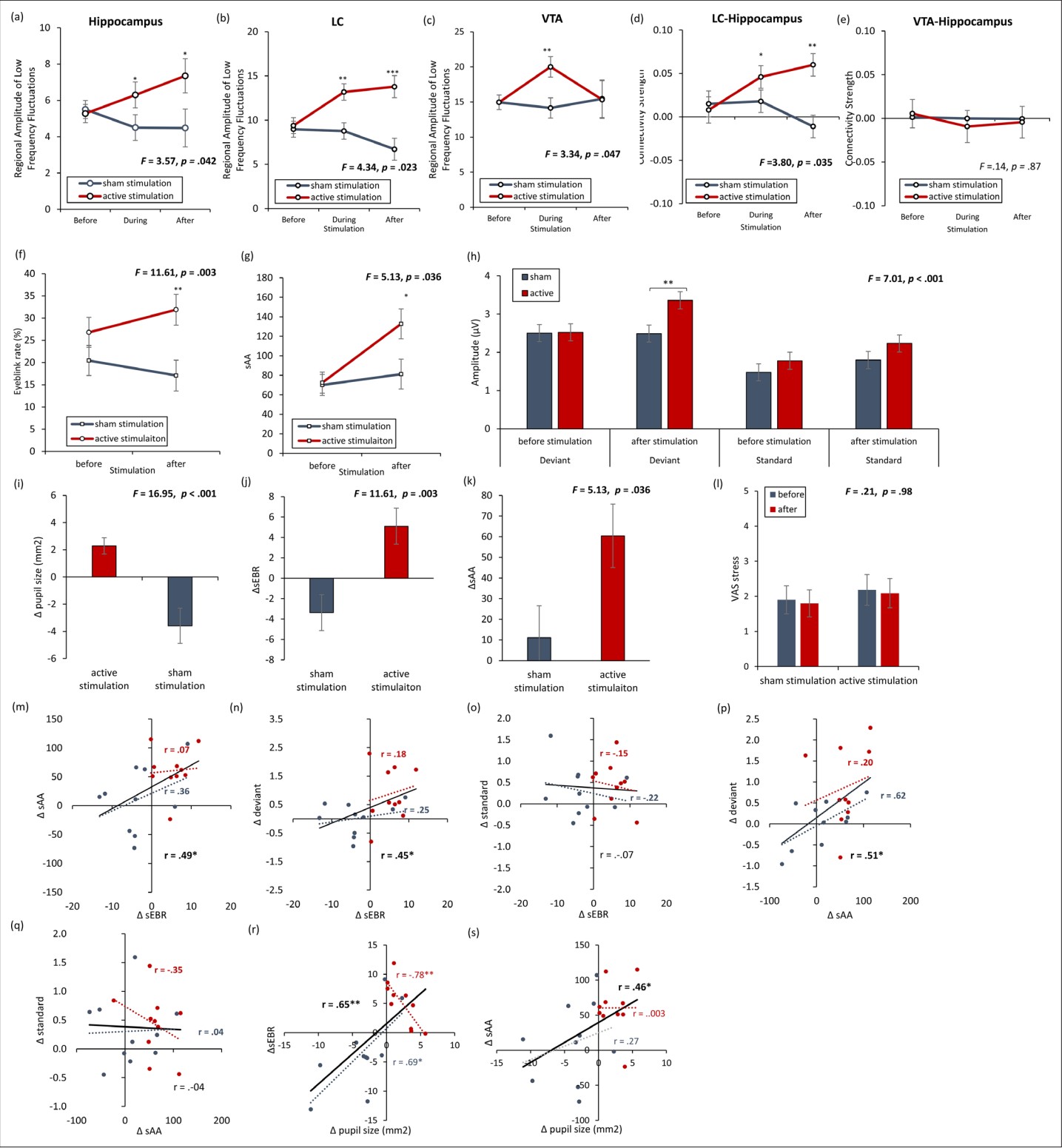

**Figure 6.** rsfMRI and physiology – locus coeruleus and dopamine with *Figure 6—figure supplement 1*. (**a, b**) The locus coeruleus and hippocampus revealed increased activity during stimulation as well as after stimulation for the active non-invasive transcutaneous electrical stimulation of the greater occipital nerve (NITESGON) group in comparison to the sham NITESGON group. (**c**) The ventral tegmental area revealed increased activity during stimulation, but not after stimulation for the active NITESGON group in comparison to the sham NITESGON group. (**d**) Increased connectivity between the locus coeruleus and hippocampus was observed during and after stimulation for the active NITESGON group in comparison to the sham NITESGON group. (**e**) No significant difference in connectivity between the ventral tegmental area and hippocampus was observed when comparing

*Figure 6 continued on next page*

*Figure 6 continued*

the active and sham NITEGSON groups during or after stimulation. (**f, g**) A significant increase in spontaneous eye blink rate and salivary α-amylase (sAA) was observed after active NITESGON in comparison to sham NITESGON. (**h**) A significant increase in peak-to-peak amplitude over the left parietal electrode side was observed for the active group in comparison to the sham group for the deviant after stimulation. (**i–l**) A significant difference when subtracting pupil size, eyeblink rates, α-amylase, and the visual analogues scale for stress before NITESGON from immediately after NITESGON showed a significant increase for pupil size, eyeblink rates, α-amylase, but not for the VAS. (**m,n, j**) A positive correlation was observed between the difference (post–pre) in spontaneous eye blink rate and the difference in sAA as well as between the difference in spontaneous eye blink rate and the difference in peak-to-peak amplitude for the deviant. (**o**) No significant correlation was observed between the difference in spontaneous eye blink rate and the difference in peak-to-peak amplitude for the standard. (**p**) A positive correlation was observed between the difference in sAA and the difference in peak-to-peak amplitude for the deviant. (**q**) No correlation was observed between the difference in sAA and the difference in peak-to-peak amplitude for the standard. (**r, s**) A positive correlation was observed between the difference (post–pre) in spontaneous eye blink rate and the difference in pupil sizeas well as between the difference in sAA and the difference pupil size. Error bars, standard error of the mean (s.e.m.). Asterisks represent significant differences (*p < 0.05; **p < 0.01; ***p < 0.001). Δ the subtraction of rate/levels before NITESGON from sAA levels immediately after NITESGON.

The online version of this article includes the following figure supplement(s) for figure 6:

**Figure supplement 1.** To verify if smoothing had an effect on the outcome, we recalculated the regional amplitude of low-frequency fluctuations without including a smoothing kernel was inspected to verify if non-invasive transcutaneous electrical stimulation of the greater occipital nerve (NITESGON) evoked activity changes in the locus coeruleus (LC), ventral tegmental area (VTA) and hippocampus.

VTA did not show changes in activity after stimulation or connectivity changes between the VTA and hippocampus during or after stimulation. However, activity changes in the VTA during NITEGSON were detected. Previous animal research has identified selective neuronal connections between the LC and VTA, implying an interaction between the LC and VTA during NITESGON may exist (*Sara, 2009*).

A key neuromodulator in memory consolidation is dopamine (DA) (*Takeuchi et al., 2016*). DA affects plasticity, synaptic transmission, and network activity in the hippocampus, and plays a critical role for hippocampal-dependent mnemonic processes by selectively enhancing consolidation of memory information (*Duszkiewicz et al., 2019*). Recent literature suggests a direct link between DA and the synaptic tag-and-capture hypothesis – the mechanism underlying behavioral tagging (*Redondo and Morris, 2011*). The core of the synaptic tag-and-capture hypothesis indicates that memory encoding creates the potential of long-term memory by creating a tag to be captured at a later stage (i.e., during memory consolidation) by protein synthesis-dependent LTP. Suggestions are made that the signal transduction processes catalyzing this synthesis of plasticity-related proteins require DA to stabilize new memories (*Morris and Frey, 1997*; *Frey and Morris, 1997*). Previous research has identified a DA agonist's ability to chemically induce LTP specifically on synapses that are activated by test stimulation, but not those that are silent (*Redondo et al., 2010*), whereas a DA antagonist reduces the memory effect 24 hr after learning (*Wang et al., 2010*), thus indicating that DA is central to the synaptic tag-and-capture hypothesis, and hence behavioral tagging (*Redondo and Morris, 2011*).

To regulate synaptic plasticity, the hippocampus receives dopaminergic input from the VTA and the LC (*Takeuchi et al., 2016*; *Sara, 2009*; *Kempadoo et al., 2016*; *Duszkiewicz et al., 2019*; *Rosen et al., 2015*; *Lisman and Grace, 2005*). However, recent research revealed that mainly LC DA mediates post-encoding memory enhancement in the hippocampus, while the VTA does not respond to arousal (i.e., novelty) (*Takeuchi et al., 2016*; *Kempadoo et al., 2016*). Animal research revealed that electrical stimulation of the LC increased DA levels and modulated hippocampal synaptic transmission (*Takeuchi et al., 2016*; *Lemon and Manahan-Vaughan, 2012*; *Devoto and Flore, 2006*). Furthermore, animal studies identified activation of the LC via optogenetic stimulation caused more LTP-related memory consolidation 45 min after stimulation (*Takeuchi et al., 2016*; *Lemon and Manahan-Vaughan, 2012*; *Devoto and Flore, 2006*). This could potentially explain why NITESGON applied while learning a memory task generated a long-term memory effect but did not modify immediate learning (*Luckey et al., 2020*; *Vanneste et al., 2020*).

A proxy for DA is spontaneous eye blink rate (sEBR), or the frequency of blinks per unit of time (*Jongkees and Colzato, 2016*). Pharmacological studies in animals and humans have shown that DA agonists elevate sEBR, whereas DA antagonist suppresses sEBR (*Jongkees and Colzato, 2016*; *Kleven and Koek, 1996*; *Karson, 1983*; *Cavanagh et al., 2014*; *Elsworth et al., 1991*; *Jutkiewicz and Bergman, 2004*; *Kaminer et al., 2011*; *Groman et al., 2014*). Moreover, sEBR is altered in clinical conditions that are associated with dysfunctions of the dopaminergic system (*Chen et al., 1996*).

sAA, pupil size as well as neurophysiology (event-related potentials, ERP) are common proxies for LC–NA activity. More specifically, pupil diameter indexes LC activity in both monkeys and humans (*Menétrey and Basbaum, 1987*), findings confirmed by intracranial recording and pharmacological challenge studies (*Kawai, 2018*). Neurophysiology utilizes the P3b ERP, which peaks at 300–600 ms after a task-relevant stimulus (*Sutton et al., 1965*; *Polich, 2007*), to indirectly measure LC–NA activity, thus presenting us with a strong cortical electrophysiological correlate of the LC–NA response (*Nieuwenhuis et al., 2005*). Using an auditory oddball task, a standard P3b-evoking task, NITESGON increased peak and mean amplitude between 300 and 600 ms immediately after stimulation for the left parietal electrode site. Therefore, we hypothesized that NITESGON would induce an increase in DA; shown via an increase in sEBR that would correlate with pupil diameter, sAA, and amplitude of the P3b after the application of NITESGON. sAA, sEBR, and ERP were collected immediately before and immediately after 20 min of NITESGON was administered.

Results showed a significant interaction effect for sEBR by condition ($F = 11.61$, p = 0.003; see *Figure 6f*), indicating that the active group (31.90 ± 10.90) had an increase in sEBR in comparison to a sham group (17.08 ± 11.07; $F = 9.10$, p = 0.007) after NITESGON. Before NITESGON no significant difference ($F = 1.77$, p = 0.20) was observed in the active group (26.80 ± 8.24) relative to the sham group (20.45 ± 12.63) in sEBR. Also, a significant interaction effect for sAA by condition ($F = 5.13$, p = 0.036; see *Figure 6g*) was obtained, revealing a significant increase in sAA ($F = 5.67$, p = 0.028) after stimulation for the active group (132.82 ± 51.23) in comparison to the sham group (81.22 ± 45.43). No significant difference ($F = 0.023$, p = 0.88) was obtained when comparing the active group (72.42 ± 43.77) versus the sham group (70.10 ± 19.93) before stimulation. Peak-to-peak amplitude analysis for P3 electrode further showed a significant effect ($F = 7.01$, p < 0.001; see *Figure 6h*). An effect was revealed between active NITESGON and sham NITESGON after stimulation ($t = 2.64$, p = 0.01). In addition, a significant effect was shown for active NITESGON after stimulation in comparison to before stimulation ($t = 2.75$, p = 0.007). A positive correlation was obtained between the difference in sEBR, and sAA (overall: $r = 0.49$, p = .029; active group: $r = 0.07$, p = 0.85; sham group: $r = 0.36$, p = 0.30; see *Figure 6m*), peak-to-peak amplitude for deviant (overall: $r = 0.45$, p = 0.048; active group: $r = 0.18$, p = 0.62; sham group: $r = 0.25$, p = 0.49; see *Figure 6n*), but noy peak-to-peak amplitude for standard (overall: $r = -0.07$, p = 0.76; active group: $r = -0.15$, p = .69; sham group: $r = -0.22$, p = 0.55; see *Figure 6o*), respectively, after NITESGON relative to before. The subtraction scores for the sEBR ($F = 11.61$, p = 0.003; *Figure 6j*) and sAA ($F = 5.13$, p = 0.036; *Figure 6k*), revealed a significant

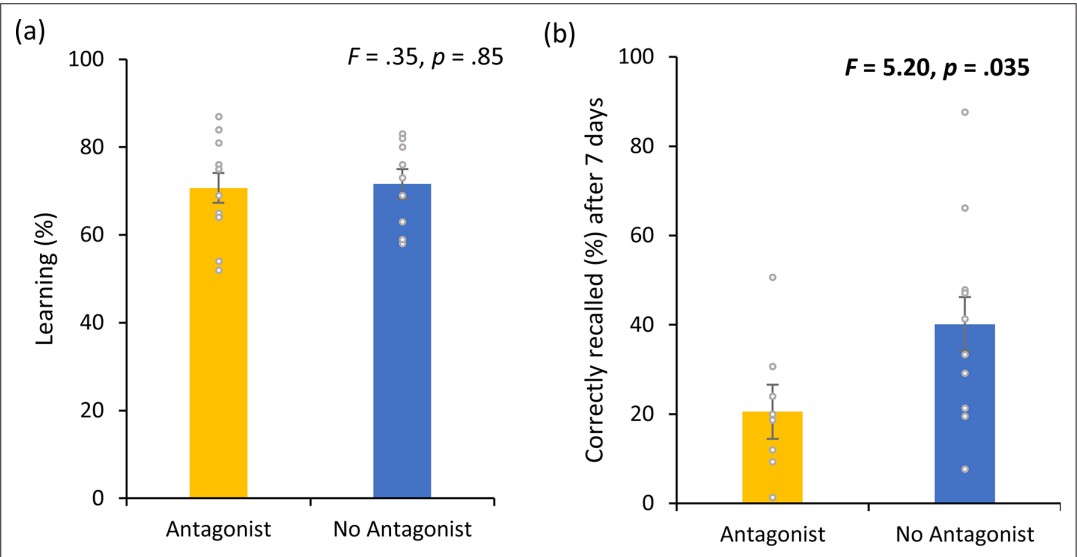

**Figure 7.** Dopamine experiment. (**a**) No difference was observed in the cumulative learning rate after non-invasive transcutaneous electrical stimulation of the greater occipital nerve (NITESGON) for participants who were taking a dopamine (DA) antagonist in comparison to participants who were not taking a DA antagonist. (**b**) A significant difference was observed in the number of recalled words after 3 or 4 days for participants who were taking a DA antagonist in comparison to participants who were not taking a DA antagonist. Error bars, standard error of the mean (s.e.m.). Asterisks represent significant differences (*p < 0.05; **p < 0.01).

difference between the active and the sham group further indicating a stimulation-induced increase in sEBR and sAA. Also, a significant correlation was obtained between sAA and peak-to-peak amplitude for deviant (overall: $r = 0.51$, p = 0.022; active group: $r = 0.20$, p = .58; sham group: $r = 0.62$, p = .054; see *Figure 6p*), but not with peak-to-peak amplitude for standard (overall: $r = -0.04$, p = 0.87; active group: $r = -0.35$, p = 0.32; sham group: $r = 0.04$, p = 0.91; see *Figure 6q*).

As previous research already showed that pupil size is an important proxy for LC mediate activity, we also look at pupil size. Our data show that a significant difference in pupil size after stimulation with baseline correction for active stimulation (2.28 ± 1.90) in comparison to sham (−3.59 ± 4.80) stimulation ($F = 16.95$, p = 0.001; see *Figure 6i*). Furthermore, a significant positive correlation was between pupil size and sEBR (overall: $r = 0.65$, p = 0.002; active group: $r = -0.78$, p = 0.008; sham group: $r = 0.69$, p = 0.029; see *Figure 6r*) and between pupil size and sAA (overall: $r = 0.46$, p = 0.042; active group: $r = 0.003$, p = 0.99; sham group: $r = 0.27$, p = 0.45; see *Figure 6s*).

## Experiment 8. Dopamine

Seeing that previous research identifies DA's vital role in memory consolidation, and NITESGON generates its effect during memory consolidation, it would be expected that blocking the DA receptor with a DA antagonist would have a direct impact on memory consolidation. To test this hypothesis, and confirm previous findings, Experiment 8 conducted the recall-only memory test 3–4 days after initial learning of the word-association task. We used the same setup as Experiment 1, whereby NITESGON was applied immediately after learning the word-association task during visit 1.

No significant effect ($F = 0.04$, p = 0.85; see *Figure 7a*) was obtained between the participants who took a DA antagonist (71.60 ± 9.18) and those who did not take a DA antagonist (70.70 ± 12.04) during the learning phase on visit 1. 7 days after learning the word associations, participants who took a DA antagonist (20.53 ± 13.32) performed worse on correctly recalling words in comparison to participants who did not take a DA antagonist (40.12 ± 23.68) ($F = 5.20$, p = 0.035; see *Figure 7b*).

## Blinding

To determine if the stimulation was well blinded, all participants in Experiments 1–7 were asked to guess if they thought they were placed in the active or control group (i.e., what stimulation participants received compared to what participants expected). Our findings demonstrated that participants

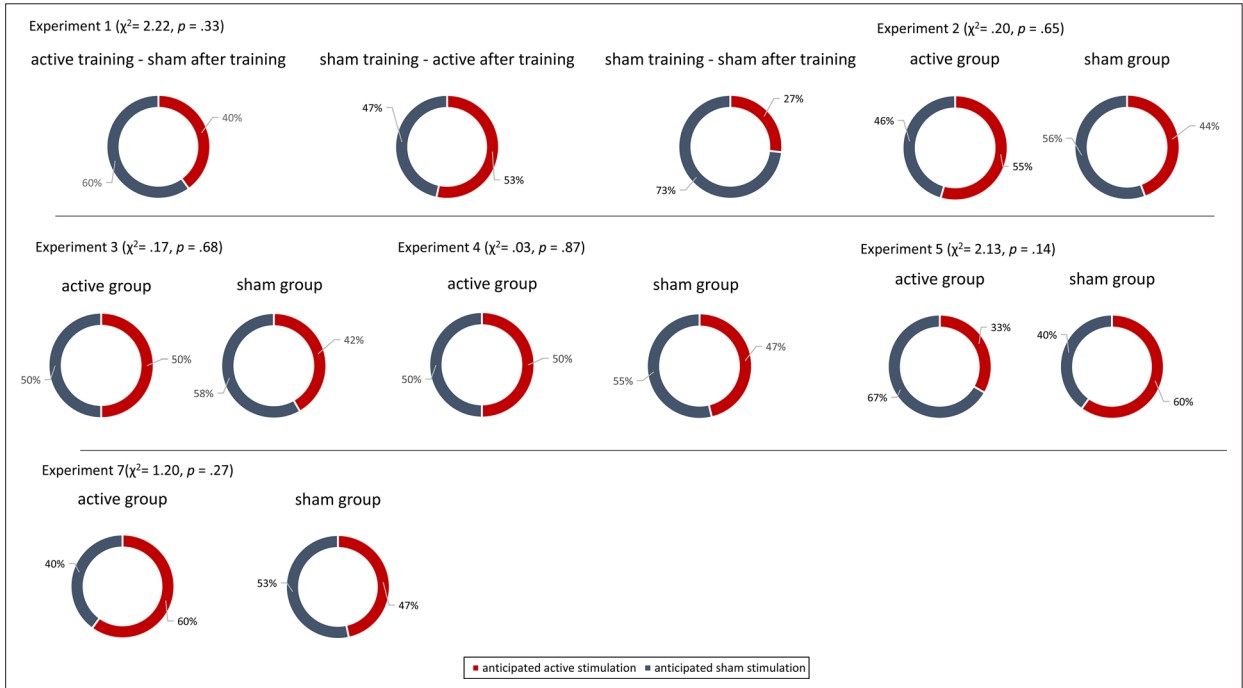

**Figure 8.** Blinding experiments. For Experiments 1–7, no difference was observed between the active and sham groups' anticipation of receiving active or sham stimulation.

could not accurately determine if they were assigned to the active or sham NITESGON group in each experiment, suggesting that our sham protocol is reliable and well blinded (see *Figure 8*).

## Discussion

Taken together, our experiments support the hypothesis that NITESGON targeting the LC strengthens hippocampal memories via the behavioral tagging mechanism. NITESGON increases activity in the LC and hippocampus during and immediately after stimulation and increases connectivity between these two areas, thus instigating initial memory consolidation and increasing the retention of memories that are formed within a window of opportunity spanning before and after LC activation. This is in accordance with the construct of behavioral tagging, which explains how memories that would normally be forgotten will endure in memory preceding, during, and following activation of the LC pathway (*Lemon and Manahan-Vaughan, 2012*; *McGaugh, 2004*; *Cahill and McGaugh, 1998*).

Notably, NITESGON does not generate an immediate memory effect during learning, however, a favorable behavioral effect is seen 7 days after initial learning. Although previous research has suggested that millisecond pairing between nerve stimulation and auditory or motor learning is essential to induce targeted plasticity (*Engineer et al., 2011*; *Marks et al., 2018*; *Conlon et al., 2020*), our data revealed that this design may not be as crucial as previously thought. This is comparable to the concept of behavioral tagging which suggests that there is a critical time window before and after training to transform a weak memory into a strong memory (*Moncada et al., 2015*). Our findings revealed that NITESGON induced a proactive and retroactive memory effect. More intriguingly, when we introduced a second word-association task (e.g., Japanese–English) that was highly anticipated to interfere with another word-association task (e.g., Swahili–English), we found that NITESGON diminished the interference effect. Interestingly, within the synaptic tag-and-capture literature, memory interference has been proposed to be the result of synaptic competition, a proposed 'fight for proteins' that arises between tagged synapses among limited proteins that leads to one memory converting to long-term memory at the expense of the other (*Okuda et al., 2021*). Therefore, it could be postulated that NITESGON could enable optimal availability of plasticity-related proteins, thus decreasing synaptic competition and subsequent interreference. However, further investigations are needed to support this postulation. Moreover, this effect induced by NITESGON did not appear to be task specific given that we saw an advantageous effect for both spatial-navigation and word-association tasks. This suggests a generalized memory enhancement effect similar to prior studies of post-encoding increases in consolidation via the LC due to inducing stress and arousal (*McGaugh, 2004*; *Cahill and McGaugh, 1998*; *Kalbe and Schwabe, 2022*).

Previous research demonstrated that LC discharge enhances synchronization of gamma activity in the hippocampus in rats (*Hajós et al., 2003*) and that gamma oscillations play an important role in long-term memories and could potentially predict subsequent recall (*Osipova et al., 2006*; *Sederberg et al., 2003*). Our results confirm this by revealing that NITESGON induces gamma changes in the medial temporal lobe that correlate with successful recall.

The role of the LC–NA system in synaptic plasticity and molecular memory consolidation has been well established over the past decades (*Sara, 2009*; *Sara, 2015*). However, recent animal studies on enhancement of memory persistence have found that LC tyrosine-hydroxylase neurons, originally defined by their canonical NA signaling, mediate post-encoding enhancement of memory in a manner consistent with possible corelease of DA from the LC axons in the hippocampus (*Takeuchi et al., 2016*; *Kempadoo et al., 2016*; *McNamara et al., 2014*). Interestingly, electrical stimulation of the LC increases DA levels that modulate hippocampal synaptic transmission up to 30 min after encoding (*Takeuchi et al., 2016*; *Lemon and Manahan-Vaughan, 2012*; *Devoto and Flore, 2006*). Our results corroborate these findings, indicating that utilizing a DA antagonist can reduce the effect of NITESGON and that sEBR, a proxy for DA, increases after NITESGON. In addition, we demonstrated changes in sAA immediately after NITESGON that correlate with memory recall 7 days later overall (including both active and sham groups). However, if we only looked at the active group, we need to be nuanced these findings as sAA levels immediately after NITESGON did not correlate with memory recall 7 days later for every experiment. Previous research indicated that α-amylase is a marker of endogenous NA activity, with human functional magnetic resonance imaging (fMRI) showing LC activity rising concomitantly with sAA levels during the viewing of emotionally arousing

slides (*van Stegeren et al., 2007*). However, original research on amylase secretion indicates that α-amylase is mediated by both DA in addition to NA (*Sundström et al., 1985*).

A prevailing hypothesis is that hippocampus-dependent memory is mediated by a subiculum–accumbens–pallidum–VTA pathway via DA (*Rossato et al., 2009*). Our results indicated that the VTA was activated during NITESGON, however, the VTA activation ceased post-stimulation. No increased connectivity was revealed between the VTA and hippocampus during or after NITESGON. This corresponds with recent research that suggests hippocampal projections in the VTA are sparse (*Takeuchi et al., 2016*; *Kempadoo et al., 2016*; *McNamara et al., 2014*) and therefore may only have a limited role in late-phase LTP (*Takeuchi et al., 2016*; *Kempadoo et al., 2016*; *McNamara et al., 2014*). However, it is possible that the VTA indirectly contributes to the formation of memories via other brain areas. Recent animal research has suggested that VTA DA neurons project to the amygdala and may contribute to fear memory in addition to the LC (*Tang et al., 2020*; *Giustino et al., 2020*), and that the VTA may contribute to synaptic consolidation independently and complementary to the LC (*Moncada, 2017*). Additionally, it is well known that the amygdala is not the final place of memory storage, but rather has major modulatory influences on the strength of a memory (*McGaugh, 2000*). Similar to the VTA in the current study, prior research has shown that the amygdala is activated during NITESGON but ceased post-stimulation; however, NITESGON was not accompanied by a task during the experiment (*Vanneste et al., 2020*). Moreover, a recent fMRI study spotlights the dynamic behavior of the LC during arousal-related memory processing stages whereby emotionally arousing stimuli triggered engagement from the LC and the amygdala during encoding; however, during consolidation and recollection stages, activity shifted to more hippocampal involvement (*Jacobs et al., 2020*). Considering the impact of the VTA and amygdala can have on memory, future experimental investigations are needed to establish their role in the memory-enhancing effects of NITESGON. One note, is that we need to be careful with the results of the LC, as this area is notoriously difficult to image due to its small size (~1–2 mm across in humans) and susceptibility to MR signal artifacts from cardiac pulsation and partial volume effects from the fourth ventricle (*Astafiev et al., 2010*). Therefore, we re-analyzed our data using no smoothing to better approximate the size of the LC, showing similar results.

Although our results reflect the putative tag-and-capture mechanism, several emotional arousal and/or noradrenergic-related frameworks may explain advantages during the consolidation phase that lead to memory enhancement effects that were observed (*Sara, 2015*; *McGaugh, 2013*; *Aston-Jones and Cohen, 2005*; *Mather and Harley, 2016*; *Nielson et al., 2005*); therefore, future research needs to be conducted to determine whether behavioral tagging explains the behavioral effects shown here. The observation supports the theory that the effect of NITESGON generated during memory consolidation were not seen immediately, but were observed 7 days later and were not affected by sleep when observed at the 12 hr mark. This is analogous to studies showing that arousal-mediated consolidation effects are time dependent but less dependent on sleep (*Park, 2005*).

When interpreting the current findings, it must be considered that some limitations exist within the research; limitations on experimental design are noted below, followed by a discussion of utilizing indirect proxy measures. The task order for Experiment 4 was randomized during the first visit for training and the recall-only memory test 7 days later; however, the word-association and spatial-navigation task used in Experiments 2 and 3 were not counterbalanced; therefore, the findings of Experiments 2 and 3 could have been impacted by a potential order effect. Additionally, in Experiment 5, the learning and test phases for the two groups were conducted at different times of day (i.e., sleep group: training at 8 p.m. and testing at 8 a.m., sleep deprivation group: training at 8 a.m. and testing at 8 p.m.), thus introducing the potential for circadian effects between the two groups. Furthermore, the recall-only memory testing occurred at the 12 hr point rather than 7 days later, allowing us to conclude that the observed effect seen 12 hr later was not affected by sleep; however, it remains unclear whether the 7-day long-term memory effects of NITESGON are sleep dependent. Moreover, it must also be acknowledged that Experiments 5 and 8 did not include a control-sham stimulation group, thus limiting the interpretation of these two experimental findings. Control-sham stimulation groups would increase our confidence in our findings that NITESGON's memory-enhancing effects depend not on sleep but on DA receptor activity.

Although the use of indirect proxy measures, such as sAA for NA activity and sEBR for DA activity, enabled the tracking of LC–NA activity changes from baseline measurements and demonstrated the potential of an LC–DA relationship, caution must be advised when interpreting results considering

these proxy measures are affiliated with limitations, such as being substantially variable, as well as the potential of other brain regions and monoamine neurotransmitters being associated with changes seen in sAA concentration levels (*Jones et al., 2020*), an enzyme that is provoked by both central parasympathetic and sympathetic nervous system activation, including acute stress responses (*Nater and Rohleder, 2009*). Additionally, although sEBR has been increasingly linked to DA, it has been defined as a more viable measure of striatal DA activity (*Jongkees and Colzato, 2016*; *Ortega et al., 2022*). At the same time, our research suggests that sEBR and DA are related as well as with the peak-to-peak amplitude for deviant and pupil size suggesting a possible common underlying mechanism. Yet, these results can only be obtained at a group level including both the active and the sham group. Lastly, it must be recognized that the effects of NITESGON obtained could be explained by activating other neuromodulatory mechanisms. Previous animal research indicates that peripheral nerve stimulation, such as vagus nerve stimulation, also activates the dopaminergic (*Brougher et al., 2021*; *Perez et al., 2014*), serotonergic (*Hulsey et al., 2019*), and cholinergic (*Hulsey et al., 2016*) pathways, all of which play an essential role in inducing long-term plasticity changes related to memory consolidation (*Avery and Krichmar, 2017*). Further studies regarding the role of additional neuromodulators would be worthwhile.

In conclusion, our work provides evidence that NITESGON is involved in the consolidation of information rather than encoding. Our findings support an implication previously put forward in the formulation of the synaptic tag-and-capture mechanism proposing late-phase LTP of synaptic activity could explain enhanced memories. As deficits in episodic memory specifically related to memory consolidation are one of the earliest detectable cognitive abnormalities in amnestic mild cognitive impairment and AD (*Yau et al., 2015*; *Weston et al., 2018*; *Wearn et al., 2020*), NITESGON might have the potential of improving memory recall by hampering the disruption of memory consolidation.

## Methods

All experiments were designed as a prospective, double-blinded, placebo-controlled, randomized parallel-group study where the researcher and the participant were blinded to the stimulation conditions. Experiment 6 was alone a single-blinded study where the participant was blinded to the stimulation condition, but not the researcher. All experiments were in accordance with the ethical standards of the Declaration of Helsinki. Experiments 1–7 were approved by the Institutional Review Board at the University of Texas at Dallas (#15-06; #17-08; #17-34; #17-84; #17-96; #18-144). All participants signed a written informed consent and consent to publish was obtained.

In all experiments, direct current was transmitted via a saline-soaked (1.3% saline) pair of synthetic sponges (5 × 7 cm) and was delivered by a specially developed, battery-driven, constant current stimulator with a maximum output of 10 mA (Eldith; http://www.neuroconn.de). For each participant receiving NITESGON, the anodal electrode was placed over the left C2 nerve dermatome and the cathodal electrode was placed over the right C2 dermatome. To maintain consistency across all participants, research assistants were trained to map out the placement according to the length of the participant's head.

To minimize skin sensations and to acclimate participants to the stimulation types, the current intensity was ramped-up (gradually increasing) until it reached its programmed maximum output (1.5 mA). After stimulating for the desired duration per group (active or sham), the current was ramped-down (gradually decreased) denoting the end of the stimulation. The impedance under each electrode was maintained under 10 kΩ. The ramp-up, ramp-down, and stimulation times were different depending on condition (active vs. sham) and experimental needs.

### Experiment 1. NITESGON during or immediately after training
#### Participants
Participants were 48 healthy, right-handed, native-English speaking adults (24 males, 24 females; mean age was 20.02 years, standard deviation [Sd] = 1.75 years) with a similar educational background (i.e., enrolled as undergraduate students at UT Dallas) with normal to corrected vision, who all had the maximum score on the Mini Mental State Examination. Participants were screened (e.g., tES contraindications, neurological impairments, not participated in a tES study) prior to enrolling into the study. None of the participants had a history of major psychiatric or neurological disorders, or any

tES contraindications, including previous history of brain injuries or epileptic insults, cardiovascular abnormalities, implanted devices, taking neuropsychiatric medications, prescribed stimulants use, or chronic use of illicit drugs (i.e., marijuana and cocaine).

Participants were excluded from the study if screening discovered they were familiar with Swahili/Arabic language or Swahili culture due to the nature of the stimuli. Furthermore, participants received instructions advising them to abstain from the following products for the associated time window prior to their study session: dental work for 48 hr, alcohol for 24 hr, caffeine and nicotine for 16 hr, and hair styling products the day of. Participants provided written, informed consent on the day of the study session.

### Word-association task

Associative memory performance was measured using a computerized Swahili–English verbal paired-associative learning task. This task was adapted from a well-established study design published in Science by *Karpicke and Roediger, 2008*. Using an $SDT_N$ paradigm (S: study phase, D: distraction phase, $T_N$: test phase with non-recalled word pairs), participants were instructed to read and remember 75-sequentially presented Swahili–English (e.g., Swahili: bustani, English: garden) word pairs made up of common day-to-day words. The Swahili–English word pairs were taken from the study by *Nelson and Dunlosky, 1994*. Participants had the opportunity to learn the list of 75-word pairs repetitively across a total of four alternating study and test periods each. During the study period, the word pairs were presented together on a computer screen for 5 s with the Swahili word on top and the English translation at the bottom ($5 \times 75 = 375$ s). The study period was followed by a cued-recall test period: Swahili cue words were presented for 8 s each during which participants had to type-in the correct English translation remembered from the study period. Correctly recalled word pairs were dropped from further testing but remained to be studied in each subsequent learning period (i.e., 4 blocks of studying 75-word pairs). The order of the words being studied or tested was randomized. Previous research has demonstrated the critical role of retrieval practice in learning of a new foreign language; therefore, the paradigm ensures that all participants were well exposed to the stimuli and avoided a ceiling effect (*Karpicke and Roediger, 2008*).

### NITESGON

There were three groups – active NITESGON during learning (i.e., study phases of the word-association task) and sham NITESGON immediately after the word-association task; sham NITESGON during learning and active NITESGON immediately the word-association task in the memory consolidation period and sham NITESGON both during and after learning of the word-association task – with 16 participants each. Active NITESGON consisted of ramp-up time of 30 s followed by a constant current of 1.5 mA (current density 0.4285 A/m²) during each of the 4 study blocks, resulting in a total stimulation time of 25 min (i.e., 375 s × 4 blocks) and ramp-down time of 30 s. For the sham NITESGON group, the current intensity was ramped-up to 1.5 mA over 30 s and immediately ramped-down over 30 s. Hence, sham NITESGON only lasted 60 s per study period, resulting in a total time of 240 s (60 s × 4 blocks) of stimulation when delivered during the study phase. For the group that received active NITESGON after the word-association task, this consisted of 30 s ramp-up and ramp-down time with 25 min of constant current stimulation at 1.5 mA. The sham NITESGON delivered after the word-association task only consisted of 30 s of ramp-up and ramp-down time resulting in 60 s of stimulation. The rationale behind the sham procedure was to mimic the transient skin sensation at the beginning of active NITESGON without producing any conditioning effects on the brain.

### Resting-state EEG

Continuous EEG data were collected from each participant pre- and post-NITESGON procedures. The data were collected using a 64-channel Neuroscan Synamps2 Quick Cap configured per the International 10–20 placement system with the midline reference located at the vertex and the ground electrode located at AFZ using the Neuroscan Scan 4.5 software (Neuroscan, http://compumedicsneuroscan.com). The impedance on each electrode was maintained at less than 5 kΩ. The data were sampled using the Neuroscan Synamps² amplifier at 500 Hz with online band-pass filtering at 0.1–100 Hz.

Eyes-closed recordings (sampling rate = 1 kHz, band passed DC–200 Hz) were obtained in a dark room which was dimly lit with a small lamp with each participant sitting upright in a comfortable chair; data collection lasted approximately 5 min. Participants were instructed not to drink alcohol 24 hr prior to EEG recording or caffeinated beverages 1 hr before recording to avoid alcohol- or caffeine-induced changes in the EEG stream. The alertness of participants was checked by monitoring both slowing of the alpha rhythm and appearance of spindles in the EEG stream to prevent possible enhancement of the theta power due to drowsiness during recording (*Moazami-Goudarzi et al., 2010*). No participants included in the current study showed such EEG changes during measurements.

## Saliva collection

Saliva was collected twice during each experiment: once immediately prior to NITESGON stimulation and once immediately after NITESGON stimulation. When the participants were ready to collect saliva, they were requested to gently tip their head backwards and collect saliva on the floor of their mouth and when ready, passively drool into the collection aid mouthpiece provided by Salimetrics laboratory (Salimetrics, LLC, USA; https://salimetrics.com). The participants were requested to collect 2 ml of saliva in one straight flow and avoid breaks between drool as much as possible. The length of time to collect 2 ml of saliva was noted and the timer was started only when participants began to passively drool into the tube. All saliva samples were stored in 2 ml cryovials, and immediately stored in an −80°C laboratory freezer. Prior to saliva collections, all participants were instructed to avoid food, sugary drinks, excess caffeine, nicotine, and acidic drinks for at least 1 hr before collecting the saliva samples. Participants were also instructed to avoid alcohol and vigorous exercise 24 hr prior to and avoid dental work 48 hr prior to their appointment. In addition, participants were instructed not to brush their teeth within 45 min of sample collection in order to avoid any risk of lowering pH levels and influencing bacterial growth. If the study was scheduled for the afternoon, participants were requested to avoid taking naps during the day. Upon completion of the collection procedures, all saliva samples were packed in dry ice and sent to the Salimetrics laboratory for analysis. We chose to use sAA as a biomarker for norepinephrine as it provided a non-invasive yet valid indicator of central sympathetic nervous system activation (*Nater and Rohleder, 2009*). sAA levels have been shown to co-vary significantly with circulating NA levels, with human fMRI showing LC activity rising simultaneously with sAA levels during viewing of emotionally arousing slides; these increases were then significantly reduced by the beta-adrenergic antagonist propranolol (*van Stegeren et al., 2007*).

## Procedure

Eligible participants were scheduled for two visits to complete the study. Visit 1 consisted of the word-association task and administration of NITESGON. Participants were randomly assigned to one of three groups during the study period. The researcher who controlled the NITESGON device was not involved in instructing the participant; this was performed by a second researcher who was blind to the stimulation protocol. rsEEG and saliva were collected twice for all participants, once immediately before and once immediately after NITESGON application. Participants were asked to refrain from studying or searching for the learned word pairs throughout the week. Participants returned 7 days after their first visit for memory testing to measure possible (long-term) effects on associative memory performance, but did not receive NITESGON, nor were they able to review word pairs. A third researcher who was not responsible for the task or NITESGON conducted the second visit.

## EEG preprocessing

For the EEG preprocessing, the data were resampled to 128 Hz, band-pass filtered (Finite Impulse Response filter) to 2–44 Hz, and re-referenced to the average reference using EEGLAB 14_1_1b (*Delorme and Makeig, 2004*). The EEG data were then plotted for a careful inspection for artifacts. All episodic artifacts suggestive of eye blinks, eye movements, jaw tension, teeth clenching, or body movements were manually removed from the EEG stream. In addition, an independent component analysis was conducted to further verify whether all artifacts were excluded.

### EEG source localization

Standardized low-resolution brain electromagnetic tomography (sLORETA) was used to estimate the intracerebral electrical sources that generated the scalp-recorded activity in each of the gamma frequency bands (30.5–44 Hz) (*Pascual-Marqui, 2002*). sLORETA computes neuronal activity as current density (A/m²) without assuming a predefined number of active sources. The sLORETA solution space consists of 6239 voxels (voxel size: 5 × 5 × 5 mm) and is restricted to cortical gray matter and hippocampi, as defined by the digitized Montreal Neurological Institute (MNI) 152 template (*Fuchs et al., 2002*). Scalpbtained in a darkes on the MNI brain are derived from the international 10–20 system (*Jurcak et al., 2007*).

The tomography of sLORETA has received considerable validation from studies combining sLORETA with other more established spatial localization methods such as fMRI (*Mulert et al., 2004*; *Vitacco et al., 2002*), structural MRI (*Worrell et al., 2000*), and positron emission tomographyon (PET) (*Dierks et al., 2000*; *Pizzagalli et al., 2004*; *Zumsteg et al., 2005*). Further sLORETA validation is based on accepting as ground truth that the localization findings obtained from invasive, implanted depth electrodes, of which there are several studies in epilepsy (*Zumsteg et al., 2006a*; *Zumsteg et al., 2006b*) and cognitive ERPs (*Volpe et al., 2007*).

### Statistics task

For visit 1 learning, a one-way analysis of variance (ANOVA) was conducted with the cumulative learning rate over the different study periods as the dependent variable and three groups as the between-subjects variable. To look at the memory effect (recall) 7 days after learning, we applied a one-way ANOVA with the group as the between-subjects variable and correctly recalled words as the dependent variable.

### Statistics saliva

Using the saliva collected via the passive drool method, sAA levels were measured. We conducted a repeated measures ANOVA with groups as between-subjects variable and sAA as within-subjects variable. A simple contrast analysis was applied to compare the different conditions using a Bonferroni correction.

### Statistics EEG – whole brain analysis

A whole brain analysis was used to compare gamma activity before and after NITESGON. These activity changes were then correlated with the number of words recalled during visit 2, 7 days after the learning task, using a Pearson correlation. Non-parametric statistical analyses of functional sLORETA images (statistical non-parametric mapping) were performed for each contrast employing a *t*-statistic for paired groups and corrected for multiple comparisons ($p < 0.05$). The significance threshold for all tests was based on a permutation test with 5000 permutations and corrected for multiple comparisons (*Nichols and Holmes, 2002*).

## Experiment 2. NITESGON during second task – retroactive strengthening of memories

### Participants

Participants were 20 healthy, right-handed, native-English speaking adults (9 males, 11 females; mean age was 21.11 years, Sd = 2.03 years) with a similar educational background (i.e., enrolled as undergraduate students at UT Dallas). Participants were screened and enrolled similar to Experiment 1.

### Word-association task (task 1)

Associative memory performance was measured using the same computerized Swahili–English verbal paired-associative learning task used in Experiment 1, however, the task consisted of three study periods in which participants were asked to read and remember 50 Swahili–English word pairs in each study period (50 × 5 = 250 s).

### Object-location task (task 2)

Participants partook in a second memory performance task immediately after the word-association task. The second memory task consisted of a spatial-navigation object-location association task

that was based on previous research (*Nilakantan et al., 2017*). Using the same $SDT_N$ paradigm, participants were instructed to view and remember 50-sequentially presented objects locations on a blue–red–gray background grid with an eye-to-screen distance of ~24 inches across three study-test blocks. The objects consisted of black and white line drawings from the Boston Naming Test; 10 objects from each of the following categories were used: animals, foods, modes of transportation, tools, and household objects. Each image appeared in a randomized order at a randomized location. Objects were presented one at a time for 5 s each (1-s interstimulus interval). Objects were presented within a white-box background (4.88 × 4.88 cm) and had a red dot superimposed at the object center to mark the precise location. Participants were instructed to study and remember the object-locations as accurately and precisely as possible. After each study phase, a cued-recall test was administered. During the test period, the studied objects were presented one at a time in the center of the screen (in a randomized order), and participants were required to recall the studied locations. At the beginning of every trial, a 2-s fixation cross at the center of the screen was presented. After this 2-s period, participants were able to use the mouse to move the object from the center of the screen to its recalled location and click a button on the mouse to indicate its final location.

The Swahili–English verbal associative task was used as task 1 and the spatial-navigation object-location task was used as task 2 for all participants.

## NITESGON

All participants received sham NITESGON during each study period of task 1 using the following parameters: a 5-s ramp-up period, followed by a constant current of 1.5 mA for 15 s, ending with a ramp-down period of 5 s, allowing for an emulated sensation of the active NITESGON. For task 2, 10 participants received active NITESGON and 10 participants received sham NITESGON. Sham stimulation parameters were the same as used in task 1 and stayed consistent in each of the three study periods of task 2. Participants given active NITESGON received a 5-s ramp-up period, followed by a constant current of 1.5 mA for 250 s, and finished with a 5-s ramp-down period during each of the three study periods of task 2 on the first day. Thus, the total simulation time for the active group was 750 s (i.e., 250-s × 3 study periods) and the sham group was 45 s (i.e., 15-s × 3 study periods). Just before the first study period of the first task participants NITESGON was delivered for 15 s to help participants habituate to the sensation and to check if they were comfortable with the sensation.

## Resting-state EEG

Continuous EEG data were collected from each participant pre- and post-NITESGON procedures as detailed in Experiment 1.

## Saliva collection

Saliva was collected twice during each experiment: once immediately prior to NITESGON stimulation and once immediately after NITESGON stimulation as detailed in Experiment 1.

## Procedure

Eligible participants were scheduled for two visits to complete the study. Visit 1 consisted of the word-association task (i.e., task 1) paired with sham stimulation, and then were randomly assigned to either the active or sham NITESGON condition for the spatial-navigation task (i.e., task 2). The researcher who controlled the NITESGON device was not involved in instructing the participant; this was performed by a second researcher who was blind to the stimulation protocol. rsEEG and saliva were collected twice for all participants, once immediately before and once immediately after NITESGON application. Participants were asked to refrain from studying or searching for the learned word pairs throughout the week. Participants returned 7 days after their first visit for memory testing on both tasks 1 and 2 to measure possible (long-term) effects on associative memory performance, but did not receive NITESGON, nor were they able to review word pairs or objects' locations. A third researcher who was not responsible for the task or NITESGON conducted the second visit.

### EEG preprocessing and source localization

The continuous EEG data were preprocessed and the source-level gamma activity pre- and post-NITESGON procedures for the two groups was determined as detailed in Experiment 1.

### Statistics task

For visit 1 learning, a multivariate analysis of variance (MANOVA) was conducted with the cumulative learning rate over the different study periods for both tasks as the dependent variable and group as the between-subjects variable. To look at the memory effect (recall) 7 days after learning, we applied an MANOVA with groups as the between-subjects variable and correctly recalled words on both tasks as the dependent variable. For both analyses, if significant, two separate one-way ANOVAs were applied with groups as the between-subjects variable and correctly recalled words as dependent variable for task 1 or 2, respectively.

### Statistics saliva

Using the saliva collected via the passive drool method, sAA levels were measured which were compared between the groups as detailed in Experiment 1.

### Statistics EEG – whole brain analysis

A whole brain analysis was used to compare gamma activity before and after NITESGON. This activity was correlated with the number of correctly recalled items (words/locations) 7 days later as detailed in Experiment 1.

## Experiment 3. NITESGON during first task – proactive strengthening of memories

### Participants

Participants were 24 healthy, right-handed, native-English speaking adults (13 males, 11 females; mean age was 20.83 years, Sd = 2.21 years) with a similar educational background (i.e., enrolled as undergraduate students at UT Dallas). Participants were screened and enrolled similar to Experiment 1.

### Word-association task (task 1)

Associative memory performance was measured using the same computerized Swahili–English verbal paired-associative learning task used in Experiment 2.

### Object-location task (task 2)

Participants partook in a second memory performance task immediately after the word-association task consisting of a spatial-navigation object-location association task used in Experiment 2.

### NITESGON

The same device, electrode placement, and active and sham NITESGON parameters described in Experiment 2 were used. Differing from Experiment 2, Experiment 3 had all participants receive active NITESGON during each study period of task 1 as opposed to task 2. Twelve participants received active NITESGON and 12 participants received sham NITESGON during the first task.

### Resting-state EEG

Continuous EEG data were collected from each participant pre- and post-NITESGON procedures as detailed in Experiment 1.

### Saliva collection

Saliva was collected twice during each experiment: once immediately prior to NITESGON stimulation and once immediately after NITESGON stimulation as detailed in Experiment 1.

## Procedure

Eligible participants were scheduled for two visits to complete the study. Visit 1 consisted of the word-association task (i.e., task 1) paired with either active or sham NITESGON, and spatial-navigation task (i.e., task 2) paired with sham NITESGON. The researcher who controlled the NITESGON device was not involved in instructing the participant; this was performed by a second researcher who was blind to the stimulation protocol. rsEEG and saliva were collected twice for all participants, once immediately before and once immediately after NITESGON application. Participants were asked to refrain from studying or searching for the learned word pairs throughout the week. Participants returned 7 days after their first visit for memory testing on both tasks 1 and 2 to measure possible (long-term) effects on associative memory performance, but did not receive NITESGON, nor were they able to review word pairs or objects' locations. A third researcher who was not responsible for the task or NITESGON conducted the second visit.

## EEG preprocessing and source localization

The continuous EEG data were preprocessed and the source-level gamma activity pre- and post-NITESGON procedures for the two groups was determined as detailed in Experiment 1.

## Statistics task

The learning in visit 1 and memory performance in visit 2 was compared between the groups as detailed in Experiment 2.

## Statistics saliva

Using the saliva collected via the passive drool method, sAA levels were measured which were compared between the groups as detailed in Experiment 1.

## Statistics EEG – whole brain analysis

A whole brain analysis was used to compare gamma activity before and after NITESGON. This activity was correlated with the number of correctly recalled items (words/locations) 7 days later as detailed in Experiment 1.

## **Experiment 4. NITESGON during interference task**

### Participants

Participants were 32 healthy, right-handed, native-English speaking adults (15 males, 17 females; mean age was 21.36 years, Sd = 2.42 years) with a similar educational background (i.e., enrolled as undergraduate students at UT Dallas). Participants were screened and enrolled similar to Experiment 1. Experiment 4 added familiarity of Japanese language or culture to the participant screening procedure, if indicated, the participant was excluded from the study due to the nature of the stimuli.

### Word-association tasks

Associative memory performance was measured using two computerized verbal paired-associate learning tasks. Similar to Experiments 2 and 3, one task comprised of the Swahili–English vocabulary learning, and the second task consisted of a newly introduced Japanese–English (e.g., Japanese: Kumo, English: cloud) word-association task. The Japanese–English word-association task used the same Swahili–English word pairs, however, the Swahili words were replaced by Japanese words.

### NITESGON

The same device, electrode placement, and active and sham NITESGON parameters described in Experiment 2 was used. Sixteen participants received active NITESGON and 16 participants received sham NITESGON during the first task, where everyone received sham NITESGON during the second task.

### Resting-state EEG

Continuous EEG data were collected from each participant pre- and post-NITESGON procedures as detailed in Experiment 1.

### Saliva collection

Saliva was collected twice during each experiment: once immediately prior to NITESGON stimulation and once immediately after NITESGON stimulation as detailed in Experiment 1.

### Procedure

Eligible participants were scheduled for two visits to complete the study. Visit 1 consisted of two word-association tasks, whereby task 1was paired with either active or sham NITESGON followed by a second word-association task (i.e., task 2) paired with sham NITESGON. The order of the two word-association tasks was randomized over the participants in a 1:1 ratio to remove a possible order effect. The researcher who controlled the NITESGON device was not involved in instructing the participant; this was performed by a second researcher who was blind to the stimulation protocol. rsEEG and saliva were collected twice for all participants, once immediately before and once immediately after NITESGON application. Participants were asked to refrain from studying or searching for the learned word pairs throughout the week. Participants returned 7 days after their first visit for memory testing on both tasks 1 and 2 to measure possible (long-term) effects on associative memory performance, but did not receive NITEGSON, nor were they able to review word pairs. A third researcher who was not responsible for the task or NITESGON conducted the second visit. As during the first visit, the two word-association tasks were randomized over the participants in a 1:1 ratio to remove a possible order effect.

### EEG preprocessing and source localization

The continuous EEG data were preprocessed and the source-level gamma activity pre- and post-NITESGON procedures for the two groups was determined as detailed in Experiment 1.

### Statistics task

For visit 1 learning, a repeated measures ANOVA was applied with the cumulative learning rate over the different study periods for both tasks as within-subjects variable and group (active vs. sham) as between-subjects variable. A similar analysis was applied for the memory effect (recall) 7 days after learning. A simple contrast analysis was applied to compare the difference conditions using a Bonferroni correction. In addition, an interference effect was determined by calculating how many words recalled during memory task 2 had an overlap from memory task 1. The higher percentage the more overlap in remembering the English translation for both exact word in tasks 1 and 2, the lesser the interference effect. A one-way ANOVA was applied with the interference effect as dependent variable and group (active vs. sham) as between-subjects variable.

### Statistics saliva

Using the saliva collected via the passive drool method, sAA levels were measured which were compared between the groups as detailed in Experiment 1.

### Statistics EEG – whole brain analysis

A whole brain analysis was used to compare gamma activity before and after NITESGON. This activity was correlated with the number of correctly recalled items (words/locations) 7 days later as detailed in Experiment 1.

## Experiment 5. Effect of NITESGON is not sleep dependent

### Participants

Participants were 20 healthy, right-handed, native-English speaking adults (11 males, 9 females; mean age was 21.18 years, Sd = 1.951 years) with a similar educational background (i.e., enrolled as undergraduate students at UT Dallas). Participants were screened and enrolled similar to Experiment 1.

### Word-association task

Associative memory performance was measured using the same computerized Swahili–English verbal paired-associative learning task used in Experiment 1.

### NITESGON

All participants received active NITESGON during each of the four study periods on visit 1 using the following parameters: a 5-s ramp-up period, followed by a constant current of 1.5 mA for 375 s (75-word pairs × 5 s), and finished with a 5-s ramp-down period. The total stimulation time was 25 min (i.e., 375 s × 4 blocks). Before the start of the first study period, an additional 15-s habituation period was added to make sure the participants got used to the sensation.

### Procedure

Eligible participants were scheduled for two visits to complete the study. Visit 1 consisted of the word-association task paired with active NITESGON. Participants were randomly assigned to one of the two groups (sleep vs. no sleep).

As Experiment 4 reveals that NITESGON is involved in synaptic consolidation that occurs over a timespan of minutes to hours after encoding the information, where sleep that has been association with systems consolidation does not play a central role. To cross validate these findings, we opted for the parallel design not including a control condition, as the aim of this study is to see if sleep mediates the memory effect induced by NITESGON, rather than the effect of NITESGON or the interaction between sleep and NITESGON on memory.

Ten participants learned the word-association task at 8:00 a.m. and were tested the same day at 8:00 p.m., while the other 10 participants learned the word-association task at 8:00 p.m. and were tested the next day at 8:00 a.m. after a night of sleep. Participants were asked to refrain from studying or searching for the learned word pairs for at least the next 12 hr. The researcher who controlled the NITESGON device was not involved in instructing the participant; this was performed by a second researcher who was blind to the stimulation protocol. A third researcher who was not responsible for the task or NITESGON conducted the second visit (12 hr later).

### Statistics task

A one-way ANOVA with group (sleep vs. no sleep) as between-subjects variable and number of words correctly recalled as dependent variable was performed.

## Experiment 6. LC – hippocampus activity and connectivity

### Participants

Participants were 32 healthy, right-handed, native-English speaking adults (16 males, 16 females; mean age was 25.32 years, Sd = 2.65 years) with a similar educational background (i.e., enrolled as undergraduate students at UT Dallas). Participants were screened and enrolled similar to Experiment 1. Experiment 6 added the exclusion of those participants who had any contraindication for MRI (i.e., metallic implants, pregnancy, claustrophobia).

### Resting-state fMRI

The resting-state fMRI data were collected on a 3T MR scanner (Achieva, Philips, Netherlands) using a 32-channel SENSE phased-array head coil. The dimensions of the coil was 38 (height) × 46 (width) × 59 (length) $cm^3$. During scanning, foam padding and earplugs were used to minimize the head movement and scanner noise. An MR-compatible, battery powered NITESGON system manufactured by MR NeuroConn Co (Germany) was applied to each participant inside the MR scanner. All the operating parts and devices that go into the scanner room were MR compatible and everything else was in the control room, connected via the waveguide. The NITESGON system was fully charged before each session and its impedance level was measured regularly to test if it was maintained at approximately 5 kΩ on each end (i.e., 10 kΩ total).

The MR session with NITESGON was divided into three consecutive blocks of scanning: before stimulation, during stimulation, and after stimulation. At the beginning of the pre-stimulation session, routine survey and T1 anatomical images were acquired for a total time of 5 min. Before acquiring the

T1 image, saline-soaked NITESGON electrodes were positioned on the subject for three consecutive blocks of rsfMRI. For each of the scanning blocks, we acquired 20-min long rsfMRI images.

For the T1 (MPRAGE) anatomical scan, parameters included a repetition time (TR) of 2300 ms, an echo time (TE) of 2.94 ms, an inversion time (TI) of 900 ms, and a flip angle of 9°. One hundred and sixty sagittal slices were taken, using a matrix size of $256 \times 256$ mm$^2$, at a $1 \times 1 \times 1$ mm$^3$ resolution.

Resting-state fMRI sequences were acquired with the following imaging parameters (echo-planar imaging protocol): TR/TE = 3000/30 ms, field-of-view $= 220 \times 220$ mm$^2$, matrix $= 64 \times 64$, number of slices $= 53$ with voxel size $= 3 \times 3 \times 4$ mm$^3$ with no gap between slices. The total number of acquired volumes was 400, counting for 20 min.

MR images were preprocessed using Statistical Parametric Mapping (SPM12b, Wellcome Department of Imaging Neuroscience, University College London, UK). Images from the first five TRs in the beginning of each session were discarded. We normalized high-resolution structural images to a standard MNI template, and segmented for three structural components, which were gray matter, white matter, and cerebrospinal fluid. Functional images were realigned to the middle volume to correct for motion artifacts. Slice-timing correction was adjusted for temporal discrepancies between $z$-direction slices acquired in an interleaved manner. After co-registration of functional volumes to the structural image (T1), volumes that contain extreme movements were linearly regressed out as covariates using Artifact Detection Tool (Gabrieli Lab, MIT, US, http://www.nitrc.org/projects/artifact_detect/). Functional images were normalized to the standard template using nonlinear transformation parameters acquired by the process of normalizing structural image to the standard template. The normalized functional images were smoothed using 6 mm Gaussian kernel.

After preprocessing, the images were processed to account for motion-related and physiological noises using an independent component analysis. Confounding factors of signals from white matter and cerebrospinal fluid were linearly regressed out from the global average brain signal using CompCor. Global signal regression was performed using the grand averaged signal from the gray matter volume. When removing confounding effect, the signal was simultaneously filtered from 0.01 to 0.17 Hz, where the maximum detectable frequency of the signal is 0.167 Hz (TR = 3 s).

## NITESGON

Shielded cables connected the MR-compatible box and electrodes, and the stimulation data were transferred via the CAT.6 LAN cable that runs throughout the MR scanner room to the non-MR-compatible stimulation devices in the control room. For the active NITESGON group, the current was ramped-up for 30 s followed by a constant current of 1.5 mA for 20 min and a 10-s ramp-down period. For sham NITESGON, the current was ramped-up over 30 s to reach the intensity of 1.5 mA followed by 15-s of constant current stimulation at 1.5 mA and 10 s ramp-down. Hence, sham NITESGON only lasted 15 s (as opposed to 20 min in the active group).

## Procedure

Participants were scanned immediately before, during, and immediately after the NITESGON stimulation. The researcher who controlled the NITESGON device was not blinded to the stimulation group, but the participant was blinded to which stimulation group they were placed in. Sixteen participants received active NITESGON and 16 received sham NITESGON.

## Statistics rsfMRI

A functional connectivity analysis was performed using the CONN toolbox. The ROI considered in the analysis were the right hippocampus (based on previous findings [*Vanneste et al., 2020*]), LC, and VTA. Both the LC and VTA were selected using probabilistic atlas (as conducted in a study across 44 adults by localizing its peak signal level in high-resolution T1 turbo spin-echo images and verified the location using post-mortem brains) (*Tona et al., 2017*). The probabilistic templates were created using processing steps specifically designed to address these difficulties (*Tona et al., 2017*). To remove potential artifacts such as head motion, respiration, and other global imaging artifacts including potential stimulation effects, we regressed out the global average brain signal.

We conducted an rALFF analysis for the LC, VTA, and hippocampus. The time series for each voxel of each ROI was transformed to the frequency domain and the power spectrum was then obtained. Since the power of a given frequency is proportional to the square of the amplitude of this frequency

component, the square root was calculated at each frequency of the power spectrum and the averaged square root was obtained across 0.01–0.17 Hz at each voxel. This averaged square root was taken as the rALFF (*Yu-Feng et al., 2007*). The rALFF of each voxel was divided by the individual global mean of the rALFF within a brain-mask, which was obtained by removing the tissues outside the brain using software MRIcron. Spatial smoothing was conducted on the maps with an isotropic Gaussian kernel of 8 mm of full width at half-maximum. A repeated measures ANOVA was used including group (active vs. sham) as between-subjects variable and rALFF before, during and after NITESGON as within-subjects variable for the different ROIs (ALFF for the VTA, LC, and hippocampus). A simple contrast analysis was included to compare the difference between active and sham stimulation for each ROI before, during, and after stimulation separately including a correction for multiple comparison (Bonferroni correction).

In addition, the average BOLD time series across all voxels within the ROIs were extracted from the smoothed functional images. A partial correlation analysis was performed, and the resulting *r*-value converted to Fisher's *Z*-transformed coefficients were used for further statistical analyses. The *Z*-transformed connectivity weights were compared between the active and sham groups for the LC and hippocampus, LC and VTA, and VTA and hippocampus, respectively, using a repeated measures ANOVA. A simple contrast analysis was applied to compare the different in active and sham condition using a Bonferroni correction.

## Experiment 7. Potential relationship between NITESGON-LC and DA

### Participants

Participants were 24 healthy, right-handed, native-English speaking adults (12 males, 12 females; mean age was 23.83 years, Sd = 2.88 years) with a similar educational background. Participants were screened and enrolled similar to Experiment 1.

### NITESGON

Active NITESGON stimulation consisted of a ramp-up period of 5 s, followed by constant current of 1.5 mA for 20 min and ramp-down period of 5 s. Sham NITESGON only consisted of a ramp-up period of 5 s to reach the intensity of 1.5 mA and an immediate ramp-down period of 5 s. Twelve participants received active NITESGON and 12 participants received sham NITESGON.

### Electrophysiological recordings

Continuous EEG data were collected from each participant in response to the auditory oddball paradigm, before and after the application of NITESGON. The auditory oddball task is a simple and well-established paradigm for the investigation of the robust P3b component which has a predictable standard tone and an unpredictable deviant tone (*Murphy et al., 2011*). The data were collected using a 64-channel Neuroscan Synamps² Quick Cap configured per the International 10–20 placement system with the midline reference located at the vertex and the ground electrode located at AFZ using the Neuroscan Scan 4.5 software. The impedance on each electrode was maintained at less than 5 kΩ. The data were sampled using the Neuroscan Synamps² amplifier at 500 Hz with online band-pass filtering at 0.1–100 Hz. Data were preprocessed using Matlab and EEGLAB in a manner similar to the original paper that showed a relationship between ERP and LC–noradrenergic arousal function (*Murphy et al., 2011*).

### Peak-to-peak P3b amplitude

Peak-to-peak amplitude was defined as the amplitude difference between the N200 peak and the P300 peak for the P3 electrode. The N200 component was identified as the most negative peak between 200 and 375 ms after the stimulus onset. The P300 component was identified as the most positive peak between 250 and 600 ms after the stimulus onset.

### Spontaneous eyeblink rate

To retain the eyeblinks, the eyeblink rate was calculated using the data before cleaning the artifacts using an independent component analysis. Furthermore, the continuous dataset before epoching was used to visualize the entire temporal profile of the eyeblink potential to avoid any cutting-off of the

potential due to epoching. An eyeblink was determined to be a sharp negative peak followed immediately by a positive peak located in the frontal electrodes such as FP1, FP2, and FPz. In some cases, the negative peak was not prominent, but the positive peak was a signatory. The topography of this potential was observed to have a clear dipole covering the frontal and frontotemporal electrodes. This potential was marked manually by a researcher, who scanned the entire EEG recording manually for all the participants in the active and sham groups, in the pre- and post-stimulation conditions. The number of eyeblinks in the length of recording was obtained and the eyeblink rate was calculated as the number of eyeblinks/minute. The same procedure was performed by a second researcher who was blinded to the conditions and the inter-rater validity was calculated. The average score was calculated from both independent researchers.

## Saliva collection
Saliva was collected twice during each experiment: once immediately prior to NITESGON stimulation and once immediately after NITESGON stimulation as detailed in Experiment 1.

## Pupil dilation
The response of the pupil to three types of light stimulation (blue, 470 nm; white 8000k color temperature; and red, 624 nm) was recorded in real time using a binocular Basler Dart Near Infrared (NIR) cameras. The lenses have a fixed focal length of 8 mm with an M12x0.5 body. The images were recorded at a frame rate of 120 Hz. There is a constant NIR illumination of the eye (850 nm) and the cameras are equipped with a 'daylight cut filter' which passes NIR and blocks any wavelengths below ~800 nm. Surface-mounted LEDs were used for light stimulation and the cameras were all mounted on a single eyepiece, which communicated with a Windows laptop through a USB 3.0 cable. Each color was shown for 200 ms first in the left eye and then in the right eye with an ISI of 8 s. This left–right trial was repeated three times for each color. The average total duration of the procedure was 2 min per participant. The participant was requested to focus on a point inside the eyepiece and open their eyes as wide as possible. They were asked to avoid rapid and frequent blinking of eyes, movement of eyes, and specifically to avoid blinking during stimulus presentation. The videos were then postprocessed to obtain the dilation of the pupil. The pupil was extracted as an ellipse from each frame with a segmentation algorithm and the diameter was calculated from the average of the major and minor axes of the resulting pupil ellipse. The difference between the maximum dilation and maximum constriction of both pupils in response to each color for every trial was calculated for each person immediately before and after NITESGON. The pupil size was measured as a direct response to the NITESGON and did not include a specific task.

## Procedure
Participants performed the auditory oddball task twice, once immediately before and once immediately after the NITESGON session. In addition, saliva and pupil dilation were also collected immediately before and immediately after the NITESGON session. The reason why we collected auditory oddball task and pupil size before and after NITESGON is to avoid a direct effect of the learning taks or inference of the current sent to the scalp. Participants were randomly assigned to the active or sham NITESGON group. The researcher who controlled the NITESGON device was not involved in instructing the participant; this was performed by a second researcher who was blind to the stimulation protocol.

## Statistics peak-to-peak P3b amplitude
EEG data were compared using a repeated measures ANOVA with groups (active vs. sham) and condition (deviant vs. standard) as the between-subjects variable, and the peak-to-peak amplitude before and after stimulation as the within-subjects variable. A simple contrast analysis was applied to compare specific effects using a Bonferroni correction.

## Statistics sEBR

We conducted a repeated measures ANOVA with group (active vs. sham) as between-subjects variable, and the average eye blink rate before and after stimulation as within-subjects variable. A simple contrast analysis was applied to compare specific contrasts using a Bonferroni correction.

## Statistics pupil size

Using the pupil size, we conducted a one-way ANOVA with groups as group variable and difference in pupil size before and after NITESGON as the dependent variable.

## Statistics saliva

Using the saliva collected via the passive drool method, sAA levels were measured which were compared between the groups as detailed in Experiment 1.

## Statistics correlation

Pearson correlations were calculated between the difference in sAA, peak-to-peak P3b amplitude and sEBR, pupil size before and after NITESGON stimulation.

## Experiment 8. Dopamine

### Participants

Participants were 20 right-handed adults (8 males, 12 females; mean age was 35.23 years, Sd = 2.63 years), half of who were selected due to their medical record indicating they were taking flupentixol (0.5 mg)/melitracen (10 mg) (Deanxit), a DA antagonist (i.e., D1 and D2) (*Hyttel, 1981*), for their tinnitus at least 2 weeks prior to the onset of the study. The remaining participants were of matching age and gender with a similar educational background. Participants were screened and enrolled similar to Experiment 1.

### Word-association task

Associative memory performance was measured using the same computerized Swahili–English verbal paired-associative learning task used in Experiment 1, however, the English words were replaced by Dutch words.

### NITESGON

All participants received active NITESGON immediately following the word-association task on visit 1 using the following parameters: a 5-s ramp-up period, followed by a constant current of 1.5 mA for 25 min, and finished with a 5-s ramp-down period.

As experiment suggests DA plays a vital role in memory consolidation, Experiment 8 is a follow-up experiment to cross validate these findings. We opted for the parallel design not including a control condition, as the aim of this study is to see if DA antagonist mediates the memory effect induced by NITESGON, rather than the effect of NITESGON or the interaction between sleep and NITESGON on memory.

### Procedure

Eligible participants were scheduled for two visits to complete the study. Visit 1 consisted of the word-association task followed by active NITESGON stimulation. Participants were asked to refrain from studying or searching for the learned word pairs throughout the week. Participants returned 3–4 days after their first visit for memory testing to measure possible (long-term) effects on associative memory performance, but did not receive NITESGON, nor were they able to review word pairs.

### Statistics task

For visit 1 learning, a one-way ANOVA was conducted with the cumulative learning rate over the different study periods as the dependent variable and two groups (antagonist or no antagonist) as between-subjects variable. To look at the memory effect (recall) 7 days after learning, we applied a

one-way ANOVA with group as the between-subjects variable and correctly recalled words as dependent variable.

## Blinding

To determine if the stimulation for all experiments was well blinded, all participants who participated in Experiments 1–7 were asked to complete a single-response questionnaire after the conclusion of the NITESGON procedure. Here, participants were asked to guess if they thought they were placed in the active or control group. A $\chi^2$ analysis was used to determine if there was a difference between what stimulation participants received compared to what participants expected.

# Additional information

### Funding

| Funder | Grant reference number | Author |
|---|---|---|
| Alzheimer Association | AARG-21-848486 | Sven Vanneste |

The funders had no role in study design, data collection, and interpretation, or the decision to submit the work for publication.

### Author contributions

Alison M Luckey, Data curation, Formal analysis, Investigation, Writing - original draft, Project administration; Lauren S McLeod, Conceptualization, Data curation, Project administration; Yuefeng Huang, Data curation, Methodology, Project administration; Anusha Mohan, Formal analysis, Writing - review and editing; Sven Vanneste, Conceptualization, Formal analysis, Supervision, Visualization, Methodology, Writing - original draft, Writing - review and editing

### Author ORCIDs

Sven Vanneste http://orcid.org/0000-0003-1513-5752

### Ethics

All experiments were in accordance with the ethical standards of the Declaration of Helsinki (1964). Experiments 1–7 were approved by the Institutional Review Board at the University of Texas at Dallas. All participants signed a written informed consent and consent to publish was obtained.

### Decision letter and Author response

Decision letter https://doi.org/10.7554/eLife.75586.sa1
Author response https://doi.org/10.7554/eLife.75586.sa2

# Additional files

### Supplementary files

• Transparent reporting form

### Data availability

Data is available: https://doi.org/10.5061/dryad.dbrv15f46.

The following dataset was generated:

| Author(s) | Year | Dataset title | Dataset URL | Database and Identifier |
|---|---|---|---|---|
| Vanneste V | 2022 | Data from: Making memories last using the peripheral effect of direct current stimulation | https://dx.doi.org/10.5061/dryad.dbrv15f46 | Dryad Digital Repository, 10.5061/dryad.dbrv15f46 |

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
