## [Editor Report]

This is a landmark study showing that non-invasive transcutaneous electrical stimulation of the greater occipital nerve (NITESGON) can improve long-term memory. The authors provide compelling evidence that applying NITESGON during learning causally influences memory behavior. This method is novel and the work presented herein will be valuable to a broad range of scientists and clinicians interested in manipulations to improve memory and cognition more broadly.

---

## [Decision Letter]

**Decision letter after peer review:**

Thank you for submitting your article "Making memories last: The peripheral effect of direct current stimulation on strengthening memories" for consideration by *eLife*. Your article has been reviewed by 2 peer reviewers, and the evaluation has been overseen by a Reviewing Editor and Floris de Lange as the Senior Editor. The following individual involved in the review of your submission has agreed to reveal their identity: Krista L Wahlstrom (Reviewer #2).

Essential revisions:

Thank you for an experimental tour-de-force! This manuscript has several strengths and the data definitely point to the result that this stimulation method and protocols do improve memory. That said, the mechanisms are oversold in the manuscript. Both reviewers agreed that several of the claims made in the paper were not warranted by the data. While the inclusion of several separate experiments is impressive and thorough, it also does limit the conclusions that can be made across experiments. A thorough and successful revision will roll back interpretational claims that are not justified by the approach or results. The critical ones to attend to are outlined clearly and thoughtfully in both reviews so I will not restate them in detail here. It is absolutely critical that the authors be upfront about the lack of specificity in their proxy measures of DA and NA activity (eyeblink and sAA). They are both indirect measures and deserve less emphasis than is currently there. In addition to addressing the reviewer comments in full, I want to highlight that the revision should address the lack of control/sham groups in Exp 5 and 8 to justify and explain how the lack of a control group changes their interpretation of the findings. The revision should better differentiate Exp 3 and 4, how do they differ? It was also noted that the interpretation of an interference effect needed to be further justified by making it clear what behavioral measure of interference was used.

*Reviewer #1 (Recommendations for the authors):*

Missing background information:

It would strengthen the paper to describe the central and autonomic pathways engaged by NITESGON. Elaborating on this neuroanatomy is important because this stimulation method doesn't target the LC specifically or directly but rather upstream targets like the vagus nerve and NTS. While the data do align with the a priori prediction that it engages learning-relevant processes in the LC, the paper would be more accessible if there was a deeper discussion of NITESGON and how it affects brainstem pathways.

Considerations for correlations:

The individual differences correlations are compelling. One key strength of the paper is how consistently some of these findings replicate. But it is also important to include separate correlations for the sham and stimulation groups. A qualitative comparison suggests that, in some cases, stimulation might decorrelate the two variables of interest (e.g., Figure 3d). This makes it difficult to interpret the linearity of the effect. Qualitatively, the between-group differences appear to obscure within-group effects, which casts some doubt on whether the relationship is linear and, importantly, remains linear in the stimulation group alone. The sample sizes are also quite small, which makes the regressions sensitive to outliers. Please report whether the correlations meet assumptions of linearity and if they are violated, adopt methods that are less sensitive to outliers (e.g., nonparametric tests).

To strengthen the argument that stimulation-induced increases in sEBR and sAA relate to oddball-evoked responses (Figure XX), it would also be useful to account for within-subject effects. Please use subtraction scores to account for these differences (Figure 6).

Spatial resolution of MRI:

The LC is notoriously difficult to image due to its small size (~1-2mm across in humans) and susceptibility to MR signal artifacts from cardiac pulsation and partial volume effects from the fourth ventricle (for a discussion, see Astiev et al., 2010). The current MRI parameters are sub-optimal for imaging the LC due to their low spatial resolution of 3 x 3 x 4 mm3 as well as the very large smoothing kernel (8mm). To help make claims about the LC, it is critical to re-analyze this data using no smoothing to better approximate the size of the LC. In its current form, the data may reflect the contribution of signals from other neighboring nuclei and brainstem regions engaged by NITESGON. In addition to performing this analysis, these issues should still be raised as important caveats in the discussion.

In addition, global signal regression in resting-state fMRI has courted much controversy and may lead to spurious correlations. It would be useful to demonstrate that the results still hold without using this preprocessing step.

It's been shown that VNS can enhance widespread effects on the BOLD signal across the brain (Cao et al., 2017). The authors observe significant increases in activation across all regions of interest in this study. To verify the specificity of stimulation effects, it would be helpful to also report the effects in at least one control region, such as the primary motor cortex. Again, VNS is a different technique than NITESGON, so more broadly it would help to differentiate whether the effects of these methods on neural function would differ. This would ground the current findings in similar prior work and highlight the advantages of using this specific method of stimulation to boost memory.

Considerations about physiological measures and interpretations:

On Page 15, it is hypothesized that sEBR would correlate with pupil diameter. However, there is no mention of pupillometry data or analysis throughout the paper. This is unfortunate because pupil diameter is perhaps the best indirect proxy of LC activity (Reimer et al., 2016; Varazzani et al., 2015; Joshi et al., 2016; Murphy et al., 2014). Is this data available? That would provide more convincing evidence of a link between learning and noradrenergic signaling.

On a related note, it has been previously shown that tVNS, which also engages the vagus nerve, elicits increases in sAA but does not affect pupil diameter (Warren et al., 2019). Furthermore, sAA is not always closely coupled with other indices of norepinephrine release (Petrakova et al., 2015). Together, these findings raise the possibility that sAA isn't a direct marker of (only) LC activation. Unfortunately, a direct relationship can't be verified in this study because LC activity and sAA weren't collected in the same study/sample. Please address the caveat that sAA might not relate directly to LC activation in the discussion.

While spontaneous blink rate has been increasingly linked to dopamine, these studies typically examine the relationship between sEBR and striatal dopamine (e.g., Jongkees and Colzato, 2016). By comparison, little work has explored the link between sEBR and hippocampal memory processes, and there is no way to explicitly link them in the current dataset. Prior work on sEBR also typically link sEBR phenomena to prefrontal executive functions (or frontostriatal loops), such as working memory or updating (Ortega et al., 2022). Thus, sEBR in effects in Experiment 7 may relate to attentional effects, as the oddball paradigm suggests, but not necessarily memory encoding. There is also some evidence that the sEBR and DA levels may be unrelated (Dang et al., 2017; Sescousse et al., 2018). While I find the current findings interesting, it's a very big stretch to interpret blink rate as reflecting LC dopamine. Like sAA, blink rate could reflect other neural processes that are engaged by NITESGON but aren't driven by a common underlying mechanism. This framing/interpretation needs to be tempered significantly. It would be more appropriate to frame the goals of that section as examining possible relationships with DA, but not LC-DA specifically. It's fair to speculate they may be related, but that is very much open to interpretation and belongs in the discussion.

Figures:

1) Correctly recall should be "correctly recalled".

2) The use of blue and red colors for both sham and stimulation conditions as well as before and after is very confusing. Please use different colors and keep consistent throughout.

3) In Figure 6i-m, keep the axes consistent (i.e., physiological measure switches from y to x for sAA and sEBR) for clarity.

4) Similarly, for Figures 3 and 4, keep A and B consistent (lines or bars) since they convey similar things.

5) The “δ deviant” etc. y-axes aren’t very descriptive. Please use clear variable names pertaining to that measure.

*Reviewer #2 (Recommendations for the authors):*

1) In line 68 “LC-NA pathway” is abbreviated before those abbreviations are written out as text.

2) Citation needed in line 95 when the authors state "as shown before…".

3) Citation needed in line 340 – "However, recent research revealed that mainly LC DA…".

4) In line 388 it would be helpful for the authors to clarify what the memory test 3 to 4 days after initial learning is and if there are results for those tests. Similarly, in Experiment 8, a justification for the 3-4 day memory test instead of the 7-day test would be helpful.

5) Typo in lines 440 and 464 – "NTESGON".

6) Line 753 – Would read easier from the perspective of the active stimulation group. i.e. "Experiment 3 had all participants receive NITESGON during each study period of task 1…".

7) Line 754 – shouldn't the active NITESGON stimulation be during the first task for this experiment?

8) It would be helpful to include individual participant data points in the bar graphs across all experiments/figures.

9) In Figure 1D there is a green sham data point in the panel legend.

[Editors' note: further revisions were suggested prior to acceptance, as described below.]

Thank you for resubmitting your work entitled "Making memories last using the peripheral effect of direct current stimulation." for further consideration by *eLife*. Your revised article has been evaluated by Floris de Lange (Senior Editor) and a Reviewing Editor.

The manuscript has been improved but there are some remaining issues that need to be addressed, as outlined below:

The reviewers both commend the authors on a very responsive revision which steps back from making causal claims that are not justified and the manuscript is more transparent now about shortcomings of the design. That said, there are some outstanding concerns that should be addressed.

The first concerns the statistical reporting of correlations that Reviewer 1 had recommended initially and still requires attention. It is important for the authors to report their correlations separately by group or to include a group as a covariate in the statistical analyses. If the correlations uncouple within a group, this should be reported and discussed.

Second, there was confusion about pupil data acquisition and when and how they were acquired. I will point your attention to Reviewer 1's specific comment here.

Third, it is important that your discussion be as specific as possible about the manipulation and the effects noted. For example Reviewer 2 points out, and correctly so, that the dependent measure of # of words recalled after stimulation may or may not reflect interference effects, as is claimed. It is recommended that the authors use a specific measure of interference if allowable, or refrain from claiming their measure(s) reflect interference.

See the reviewers' specific comments below:

*Reviewer #1 (Recommendations for the authors):*

First, I commend the authors on the substantial work they invested in this revision. They have addressed most of my comments and this was no small effort. I especially appreciate that they have acknowledged the lack of causality and direct associations between neuromodulators and some of these indirect biomarkers. However, I do have a few outstanding concerns that I believe warrant consideration prior to publication.

1) The pupil analyses were a little puzzling. It would more sensible to measure discrete pupil dilations to the deviant vs. standard stimuli, as LC responses are most often measured to reflect "cognitive activity" rather than luminance. The LC will be stimulated by blue light, but it would be stronger to report oddball-evoked effects. I'm not sure if I misunderstood whether the pupil wasn't recorded during the task itself.

2) I'd still recommend reporting separate correlations by group. Many of the correlation effects appear to uncouple in the stimulation condition, which could alter some interpretations of the data. If this happens, additional discussion is warranted.

3) The addition of the "before" and "after" (blue/yellow colors) in some of the keys was a useful change. However, these names could be more descriptive. It's not readily apparent what these mean when the other variable also describes timepoints (e.g., during stimulation).

4) There are quite a number of grammatical and spelling errors throughout, which I won't point out individually. The paper should be proofread again and revised before final acceptance.

5) It would be useful to foreground the "competition" effects of tag-and-capture in the Results section for the interference studies. This is described well in the discussion, but the same point should be repeated higher up. Otherwise, this buries the lead for the reader.

---

## [Author Response]

Essential revisions:Thank you for an experimental tour-de-force! This manuscript has several strengths and the data definitely point to the result that this stimulation method and protocols do improve memory. That said, the mechanisms are oversold in the manuscript. Both reviewers agreed that several of the claims made in the paper were not warranted by the data. While the inclusion of several separate experiments is impressive and thorough, it also does limit the conclusions that can be made across experiments. A thorough and successful revision will roll back interpretational claims that are not justified by the approach or results. The critical ones to attend to are outlined clearly and thoughtfully in both reviews so I will not restate them in detail here.

Thank you for an experimental tour-de-force! This manuscript has several strengths, and the data definitely point to the result that this stimulation method and protocols do improve memory. That said, the mechanisms are oversold in the manuscript. Both reviewers agreed that several of the claims made in the paper were not warranted by the data. While the inclusion of several separate experiments is impressive and thorough, it also does limit the conclusions that can be made across experiments. A thorough and successful revision will roll back interpretational claims that are not justified by the approach or results. The critical ones to attend are outlined clearly and thoughtfully in both reviews so I will not restate them in detail here.

It is absolutely critical that the authors be upfront about the lack of specificity in their proxy measures of DA and NA activity (eyeblink and sAA). They are both indirect measures and deserve less emphasis than is currently there.

Upon consideration of your comment, we have included a paragraph in the Discussion section highlighting limitations that exist within the research. Within this paragraph, we have emphasized that the indirect proxy measures of sAA and sEBR are associated with being substantially variable and that sAA may also reflect changes in monoamine neurotransmitters of other brain regions. Here we advise that readers take caution when interpreting the results due to these limitations and lack of specificity in the proxy measures.

Please see the newly added text from the Discussion section on page 22 below:

“Although the use of indirect proxy measures, such as sAA for NA activity and sEBR for DA activity, enabled the tracking of LC-NA activity changes from baseline measurements and demonstrated the potential of an LC-DA relationship, caution must be advised when interpreting results considering these proxy measures are affiliated with limitations, such as being substantially variable, as well as the potential of other brain regions and monoamine neurotransmitters being associated with changes seen in sAA concentration levels(82), an enzyme that is provoked by both central parasympathetic and sympathetic nervous system activation, including acute stress responses(83). Additionally, although sEBR has been increasingly linked to DA, it has been defined as a more viable measure of striatal DA activity (54, 84). At the same time, our research suggests that sEBR and DA are related as well as with the peak-to-peak amplitude for deviant suggesting at least a possible common underlying mechanism. Lastly, it must be recognized that the effects of NITESGON obtained could be explained by activating other neuromodulatory mechanisms. Previous animal research indicates that peripheral nerve stimulation, such as vagus nerve stimulation, also activates the dopaminergic (87, 88), serotonergic (89), and cholinergic (90) pathways, all of which play an essential role in inducing long-term plasticity changes related to memory consolidation(91). Further studies regarding the role of additional neuromodulators would be worthwhile.”

In addition to addressing the reviewer comments in full, I want to highlight that the revision should address the lack of control/sham groups in Exp 5 and 8 to justify and explain how the lack of a control group changes their interpretation of the findings.

Upon review of your comment, we have addressed the lack of control/sham groups in Experiment 5 and 8 in the Discussion section when acknowledging the limitations of the research. Additionally, we have pointed out that including control/sham groups in these experiments would influence our interpretation of the results by increasing our confidence of NITESGON’s memory-enhancing effect not being sleep dependent but rather being dependent upon DA receptor activity.

Please see the newly added text from the Discussion section on pages 21-22 below:

“Furthermore, the recall-only memory testing occurred at the 12-hour point rather than 7-days later, allowing us to conclude that the observed effect seen 12-hours later was not affected by sleep; however, it remains unclear whether the 7-day long-term memory effects of NITESGON are sleep-dependent. Moreover, it must also be acknowledged that Experiments 5 and 8 did not include a control-sham stimulation group, thus limiting the interpretation of these two experimental findings. Control-sham stimulation groups would increase our confidence in our findings that NITESGON’s memory-enhancing effects depend not on sleep but on DA receptor activity.”

The revision should better differentiate Exp 3 and 4, how do they differ?

In order to clarify what the differences were between Experiment 3 and Experiment 4, we added text in the paragraph introducing Experiment 4 clarifying that the object-location task used in Experiment 3 was replaced in Experiment 4 with a Japanese-English verbal associative learning task.

Please see the paragraph from the Experiment 4 subsection on page 10 below:

“Experiments 2 and 3 revealed both retroactive and proactive memory effects 7-days after initial learning of the two tasks. To further explore if NITESGON is linked to behavioral tagging and evaluate if interference impacts NITESGON as the strong stimulus, Experiment 4 removed the object-location task used in Experiments 2 and 3 and replaced it with a Japanese-English verbal associative learning task similar to the Swahili-English verbal associative task. Considering how memory formation and persistence are susceptible to interference occurring pre-and post-encoding(37-39) and are heavily influenced by commonality amongst the learned and intervening stimuli(40); it is believed that conducting two consecutive, like-minded word-association (i.e., Swahili-English and Japanese-English) tasks will result in one’s consolidation process interfering with that of the other(41). Considering how our previous experiments suggest the effect obtained by NITESGON improves the consolidation of information via behavioral tagging, it is possible that NITESGON on the first task might help reduce the overall interference effect on the second task.”

It was also noted that the interpretation of an interference effect needed to be further justified by making it clear what behavioral measure of interference was used.

Upon review of this comment, we thought that it was best to modify the paper in two ways. First, we explained in further detail that comparing the percentage of correctly recalled word pairs on the second task 7-days after learning from the percentage of correctly recalled word pairs on the first task 7-days after learning was done to measure for an interference effect.

Please see the adapted text from the Experiment 4 subsection on page 11 below:

“Upon assessment for a potential interference effect, the active group displayed no significant difference in how many words participants were able to recall between the first and the second task (difference:.76 ±4.93) (F = .29, p = .60), whereas the sham group demonstrated the first task rendered an interference effect on the second task (difference: 5.16 ±5.99) (F = 14.11, p = .001).”

Additionally, the methods section describing how the interference effect was calculated was changed. The newly edited text better explains that the percentage of words pairs learned were subtracted from one another to measure the significance of interference one may have potentially had on the other.

Please see the amended text in the Methods section on page 38 below:

“In addition, an interference effect was calculated by subtracting the percentage of correctly recalled word pairs on the second task 7-days after learning from the percentage of correctly recalled word pairs on the first task 7-days after learning. This number gave a proxy of interference.”

Reviewer #1 (Recommendations for the authors):Missing background information:It would strengthen the paper to describe the central and autonomic pathways engaged by NITESGON. Elaborating on this neuroanatomy is important because this stimulation method doesn't target the LC specifically or directly but rather upstream targets like the vagus nerve and NTS. While the data do align with the a priori prediction that it engages learning-relevant processes in the LC, the paper would be more accessible if there was a deeper discussion of NITESGON and how it affects brainstem pathways.

Per your suggestion, we have provided a description of the potential pathway stimulation may take to activate the LC-NA system.

Please see the newly added text from the Introduction section on page 3 below:

“In addition, our recent work suggests that non-invasive transcutaneous electrical stimulation of the greater occipital nerve (NITESGON) using direct current utilizes a pathway that arises from the C2 spinal nerve to establish communication gateways from the periphery to the brain via afferent fibers that project to the brainstem and synapse onto the nucleus tractus solitarius (NTS); various rodent studies have used anterograde neuronal tracers to demonstrate these connections (8-11). From the NTS, the information is then integrated amongst networks within the complex reticular formation and relayed across the brainstem through major cortical and subcortical regions(12).”

We have also acknowledged how the effects of NITESGON may be explained via the activation of other neuromodulatory pathways via peripheral nerve stimulation. To elaborate, we draw parallels with how vagus nerve stimulation can activate the dopaminergic, serotonergic, and cholinergic pathways, all of which play a role in plasticity changes.

Please see the newly added text from the Discussion section on page 22 below:

“Lastly, it must be recognized that the effects of NITESGON obtained could be explained by activating other neuromodulatory mechanisms. Previous animal research indicates that peripheral nerve stimulation, such as vagus nerve stimulation, also activates the dopaminergic(85), serotonergic(86), and cholinergic(87) pathways, all of which play an essential role in inducing long-term plasticity changes related to memory consolidation(88). Further studies regarding the role of additional neuromodulators would be worthwhile.”

Considerations for correlations:The individual differences correlations are compelling. One key strength of the paper is how consistently some of these findings replicate. But it is also important to include separate correlations for the sham and stimulation groups. A qualitative comparison suggests that, in some cases, stimulation might decorrelate the two variables of interest (e.g., Figure 3d). This makes it difficult to interpret the linearity of the effect. Qualitatively, the between-group differences appear to obscure within-group effects, which casts some doubt on whether the relationship is linear and, importantly, remains linear in the stimulation group alone. The sample sizes are also quite small, which makes the regressions sensitive to outliers. Please report whether the correlations meet assumptions of linearity and if they are violated, adopt methods that are less sensitive to outliers (e.g., nonparametric tests).

We understand the concern and included for correlation the Spearman rank in addition to the Pearson to get a better idea of all correlations. We implemented a supplement figure just the ranks to present the Spearman correlation. See supl Figure 1 and text.

To strengthen the argument that stimulation-induced increases in sEBR and sAA relate to oddball-evoked responses (Figure XX), it would also be useful to account for within-subject effects. Please use subtraction scores to account for these differences (Figure 6).

We included the subtraction score for both sEBR and sAA in the paper.

“The subtraction scores for the sEBR (F = 11.61, p = .003; see Figure 3) and sAA (F = 5.13, p = .036; see Figure 3), revealed a significant difference between the active and the sham group further indicatin a stimulation-induced increases in sEBR and sAA.”

Spatial resolution of MRI:The LC is notoriously difficult to image due to its small size (~1-2mm across in humans) and susceptibility to MR signal artifacts from cardiac pulsation and partial volume effects from the fourth ventricle (for a discussion, see Astiev et al., 2010). The current MRI parameters are sub-optimal for imaging the LC due to their low spatial resolution of 3 x 3 x 4 mm3 as well as the very large smoothing kernel (8mm). To help make claims about the LC, it is critical to re-analyze this data using no smoothing to better approximate the size of the LC. In its current form, the data may reflect the contribution of signals from other neighboring nuclei and brainstem regions engaged by NITESGON. In addition to performing this analysis, these issues should still be raised as important caveats in the discussion.In addition, global signal regression in resting-state fMRI has courted much controversy and may lead to spurious correlations. It would be useful to demonstrate that the results still hold without using this preprocessing step.

We understand the concern and included an additional analysis not using a smoothing kernel. Our result remains more or less the same. We included this additional analysis in supplement material and at this to results and Discussion section.

“To further confirm our data, we replicated our analysis by not including a smoothing kernel, showing similar results for the LC, VTA and hippocampus (see figure 6 suppl. Figure 2). Furthermore, we included two control areas, the left inferior parietal cortex, where we do not expect to see any changes (see figure 6 suppl Figure 2).”

“One note, is that we need to be careful with the results of the LC, as this area is notoriously difficult to image due to its small size (~1-2mm across in humans) and susceptibility to MR signal artifacts from cardiac pulsation and partial volume effects from the fourth ventricle (78). Therefore, we re-analyzed our data using no smoothing to better approximate the size of the LC, showing similar results.”

It's been shown that VNS can enhance widespread effects on the BOLD signal across the brain (Cao et al., 2017). The authors observe significant increases in activation across all regions of interest in this study. To verify the specificity of stimulation effects, it would be helpful to also report the effects in at least one control region, such as the primary motor cortex. Again, VNS is a different technique than NITESGON, so more broadly it would help to differentiate whether the effects of these methods on neural function would differ. This would ground the current findings in similar prior work and highlight the advantages of using this specific method of stimulation to boost memory.

We include a different area not showing effect.

Considerations about physiological measures and interpretations:On Page 15, it is hypothesized that sEBR would correlate with pupil diameter. However, there is no mention of pupillometry data or analysis throughout the paper. This is unfortunate because pupil diameter is perhaps the best indirect proxy of LC activity (Reimer et al., 2016; Varazzani et al., 2015; Joshi et al., 2016; Murphy et al., 2014). Is this data available? That would provide more convincing evidence of a link between learning and noradrenergic signaling.

We included these findings in both results and method section.

“Experiment 7. Potential relationship between NITESGON-LC and dopamine.

The previous experiment revealed activity changes in both the LC and hippocampus as well as increased connectivity between the LC and hippocampus both during and after NITESGON. Conversely, the VTA did not show changes in activity after stimulation or connectivity changes between the VTA and hippocampus during or after stimulation. However, activity changes in the VTA during NITEGSON were detected. Previous animal research has identified selective neuronal connections between the LC and VTA, implying an interaction between the LC and VTA during NITESGON may exist(31).

[…]

As previous research already showed that pupil size is an important proxy for LC mediate activity, we also look at pupil size. Our data show that a significant difference in pupil size after stimulation with baseline correction for active stimulation (2.28 +1.90) in comparison to sham (-3.59 +4.80) stimulation (F = 16.95, p = .001; see Figure 6i). Furthermore, a significant positive correlation was between pupil size and sEBR (r = .65, p = .002; see Figure 6r) and between pupil size and sAA (r = .46, p = .042; see Figure 6s).”

On a related note, it has been previously shown that tVNS, which also engages the vagus nerve, elicits increases in sAA but does not affect pupil diameter (Warren et al., 2019). Furthermore, sAA is not always closely coupled with other indices of norepinephrine release (Petrakova et al., 2015). Together, these findings raise the possibility that sAA isn't a direct marker of (only) LC activation. Unfortunately, a direct relationship can't be verified in this study because LC activity and sAA weren't collected in the same study/sample. Please address the caveat that sAA might not relate directly to LC activation in the discussion.

Upon consideration of your comment, we have included a paragraph in the Discussion section highlighting limitations that exist within the research. Within this paragraph, we have emphasized that the indirect proxy measures of sAA is associated with being substantially variable and that sAA may also reflect changes in monoamine neurotransmitters of other brain regions. Here we advise that readers take caution when interpreting the results due to these limitations and lack of specificity in using sAA as a proxy measure.

Please see the newly added text from the Discussion section on page 22 below:

“Although the use of indirect proxy measures, such as sAA for NA activity and sEBR for DA activity, enabled the tracking of LC-NA activity changes from baseline measurements and demonstrated the potential of an LC-DA relationship, caution must be advised when interpreting results considering these proxy measures are affiliated with limitations, such as being substantially variable, as well as the potential of other brain regions and monoamine neurotransmitters being associated with changes seen in sAA concentration levels(80), an enzyme that is provoked by both central parasympathetic and sympathetic nervous system activation, including acute stress responses(81).”

While spontaneous blink rate has been increasingly linked to dopamine, these studies typically examine the relationship between sEBR and striatal dopamine (e.g., Jongkees and Colzato, 2016). By comparison, little work has explored the link between sEBR and hippocampal memory processes, and there is no way to explicitly link them in the current dataset. Prior work on sEBR also typically link sEBR phenomena to prefrontal executive functions (or frontostriatal loops), such as working memory or updating (Ortega et al., 2022). Thus, sEBR in effects in Experiment 7 may relate to attentional effects, as the oddball paradigm suggests, but not necessarily memory encoding. There is also some evidence that the sEBR and DA levels may be unrelated (Dang et al., 2017; Sescousse et al., 2018). While I find the current findings interesting, it's a very big stretch to interpret blink rate as reflecting LC dopamine. Like sAA, blink rate could reflect other neural processes that are engaged by NITESGON but aren't driven by a common underlying mechanism. This framing/interpretation needs to be tempered significantly. It would be more appropriate to frame the goals of that section as examining possible relationships with DA, but not LC-DA specifically. It's fair to speculate they may be related, but that is very much open to interpretation and belongs in the discussion.

Upon consideration of your comment, we have included a paragraph in the Discussion section highlighting limitations that exist within the research. Within this paragraph, we have emphasized that the indirect proxy measures of sEBR is associated with being substantially variable. Here we advise that readers take caution when interpreting the results due to these limitations and lack of specificity in the proxy measures.

Please see the newly added text from the Discussion section on page 22 below:

“Additionally, although sEBR has been increasingly linked to DA, it has been defined as a more viable measure of striatal DA activity(52, 82). At the same time, some evidence suggests that sEBR and DA levels may be unrelated(83, 84), thus requiring further validation as a behavioral proxy measure.”

Figures:1) Correctly recall should be "correctly recalled".

We changed this in all figures.

2) The use of blue and red colors for both sham and stimulation conditions as well as before and after is very confusing. Please use different colors and keep consistent throughout.

We change this.

3) In Figure 6i-m, keep the axes consistent (i.e., physiological measure switches from y to x for sAA and sEBR) for clarity.

We included these changes.

4) Similarly, for Figures 3 and 4, keep A and B consistent (lines or bars) since they convey similar things.

We made sure that it is consistent.

5) The "δ deviant" etc. y-axes aren't very descriptive. Please use clear variable names pertaining to that measure.

We explain this in the figure legends.

Reviewer #2 (Recommendations for the authors):1) In line 68 "LC-NA pathway" is abbreviated before those abbreviations are written out as text.

Thank you for spotting our mistake in abbreviating “LC-NA pathway” before we wrote out the abbreviations. We have edited the text so that “LC-NA pathway” is now written out and the next appearance of the term is abbreviated.

Please see the corrected text on page 4 below:

“The neural mechanism that controls this novelty response is the locus coeruleus-noradrenaline (LC-NA) pathway(24, 25). Animal research further indicates that direct electrical stimulation of the locus coeruleus (LC) modulates hippocampal synaptic consolidation(26-28).”

2) Citation needed in line 95 when the authors state "as shown before…".

Thank you for noticing a citation was missed and bringing it to our attention. We have edited the text so that the proper citation is included.

Please see the citation added on page 5 below:

“Seeing that NITESGON activates the LC pathway, which plays an important role in memory consolidation, we hypothesize that participants will be able to establish long-term memories upon modulating the LC both during learning, as shown before(13, 14), as well as immediately after learning.”

3) Citation needed in line 340 – "However, recent research revealed that mainly LC DA…".

Thank you for noticing a citation was missed and bringing it to our attention. We have edited the text so that the proper citation is included.

Please see the citation added on page 15 below:

“However, recent research revealed that mainly LC DA mediates post-encoding memory enhancement in the hippocampus, while the VTA does not respond to arousal (i.e., novelty)(25, 43).”

4) In line 388 it would be helpful for the authors to clarify what the memory test 3 to 4 days after initial learning is and if there are results for those tests. Similarly, in Experiment 8, a justification for the 3-4 day memory test instead of the 7-day test would be helpful.

Upon review of the paper and your comment, we have added clarification to the sentence by indicating that the memory test 3 to 4 days after initial learning was the recall-only memory test. The justification for including the memory test 3-4 days after initial learning rather than 7 days after initial learning lies within participants of Experiment 8 falling in an older age population. The 3-to-4-day separation between initial learning and memory testing allowed for a fixed scheduling practice and enabled a more convenient scheduling policy that was easier for the population to subscribe to.

Please see the amended text from page 17 below:

“To test this hypothesis, and confirm previous findings, Experiment 8 conducted the recall-only memory test 3 to 4-days after initial learning of the word-association task.”

5) Typo in lines 440 and 464 – "NTESGON".

Thank you for catching the spelling mistakes on line 440 and line 464 and bringing them to our attention. We have reviewed the spelling errors and have corrected them to “NITESGON.”

Please see the corrected text on pages 19 and 22 below:

“In addition, we demonstrated changes in sAA immediately after NITESGON that correlate with memory recall 7-days later.”

“In conclusion, our work provides evidence that NITESGON is involved in the consolidation of information rather than encoding.”

6) Line 753 – Would read easier from the perspective of the active stimulation group. i.e. "Experiment 3 had all participants receive NITESGON during each study period of task 1…".

Per your suggestion, we have edited the wording so that it is described from the perspective of the active stimulation group and is easier for the reader to understand.

Please see the new wording on page 34 below:

“Differing from Experiment 2, Experiment 3 had all participants receive active NITESGON during each study period of task 1 as opposed to task 2.”

7) Line 754 – shouldn't the active NITESGON stimulation be during the first task for this experiment?

Thank you for spotting and pointing out the error on line 754. We have edited the paper so that line 754 has been corrected and states that NITESGON stimulation was administered during the first task.

Please see the corrected text on page 34 below:

“12 participants received active NITESGON and 12 participants received sham NITESGON during the first task.”

8) It would be helpful to include individual participant data points in the bar graphs across all experiments/figures.

We include this in the figures.

9) In Figure 1D there is a green sham data point in the panel legend.

We resolve this issue.

[Editors' note: further revisions were suggested prior to acceptance, as described below.]

The manuscript has been improved but there are some remaining issues that need to be addressed, as outlined below:The reviewers both commend the authors on a very responsive revision which steps back from making causal claims that are not justified and the manuscript is more transparent now about shortcomings of the design. That said, there are some outstanding concerns that should be addressed.The first concerns the statistical reporting of correlations that Reviewer 1 had recommended initially and still requires attention. It is important for the authors to report their correlations separately by group or to include a group as a covariate in the statistical analyses. If the correlations uncouple within a group, this should be reported and discussed.

We added the correlations for each group separately for each of the experiments. A description was included for each experiment in the Results section. All figures include also the correlation and regression line for each group separately.

Second, there was confusion about pupil data acquisition and when and how they were acquired. I will point your attention to Reviewer 1's specific comment here.

We clarify this in the paper. “Pupil dilation. The response of the pupil to three types of light stimulation (blue, 470 nm; white 8000k color temperature; and red, 624 nm) was recorded in real-time using a binocular Basler Dart Near Infrared (NIR) cameras. The lenses have a fixed focal length of 8 mm with an M12x0.5 body. The images were recorded at a frame rate of 120 Hz. There is a constant NIR illumination of the eye (850 nm) and the cameras are equipped with a “daylight cut filter” which passes NIR and blocks any wavelengths below ~800 nm. Surface-mounted LEDs were used for light stimulation and the cameras were all mounted on a single eyepiece, which communicated with a Windows laptop through a USB 3.0 cable. Each color was shown for 200 ms first in the left eye and then in the right eye with an ISI of 8 seconds. This left-right trial was repeated three times for each color. The average total duration of the procedure was two minutes per participant. The participant was requested to focus on a point inside the eyepiece and open their eyes as wide as possible. They were asked to avoid rapid and frequent blinking of eyes, movement of eyes, and specifically to avoid blinking during stimulus presentation. The videos were then post-processed to obtain the dilation of the pupil. The pupil was extracted as an ellipse from each frame with a segmentation algorithm and the diameter was calculated from the average of the major and minor axes of the resulting pupil ellipse. The difference between the maximum dilation and maximum constriction of both pupils in response to each color for every trial was calculated for each person immediately before and after NITESGON. The pupil size was measured as a direct response to the NITESGON and did not include a specific task.”

Third, it is important that your discussion be as specific as possible about the manipulation and the effects noted. For example Reviewer 2 points out, and correctly so, that the dependent measure of # of words recalled after stimulation may or may not reflect interference effects, as is claimed. It is recommended that the authors use a specific measure of interference if allowable, or refrain from claiming their measure(s) reflect interference.

We understand the concern and apologies for the confusion. We recalculated the interference effect as suggested by reviewer 2.

See the reviewers' specific comments below:Reviewer #1 (Recommendations for the authors):First, I commend the authors on the substantial work they invested in this revision. They have addressed most of my comments and this was no small effort. I especially appreciate that they have acknowledged the lack of causality and direct associations between neuromodulators and some of these indirect biomarkers. However, I do have a few outstanding concerns that I believe warrant consideration prior to publication.1) The pupil analyses were a little puzzling. It would more sensible to measure discrete pupil dilations to the deviant vs. standard stimuli, as LC responses are most often measured to reflect "cognitive activity" rather than luminance. The LC will be stimulated by blue light, but it would be stronger to report oddball-evoked effects. I'm not sure if I misunderstood whether the pupil wasn't recorded during the task itself.

We clarify this in the paper. “Pupil dilation. The response of the pupil to three types of light stimulation (blue, 470 nm; white 8000k color temperature; and red, 624 nm) was recorded in real-time using a binocular Basler Dart Near Infrared (NIR) cameras. The lenses have a fixed focal length of 8 mm with an M12x0.5 body. The images were recorded at a frame rate of 120 Hz. There is a constant NIR illumination of the eye (850 nm) and the cameras are equipped with a “daylight cut filter” which passes NIR and blocks any wavelengths below ~800 nm. Surface-mounted LEDs were used for light stimulation and the cameras were all mounted on a single eyepiece, which communicated with a Windows laptop through a USB 3.0 cable. Each color was shown for 200 ms first in the left eye and then in the right eye with an ISI of 8 seconds. This left-right trial was repeated three times for each color. The average total duration of the procedure was two minutes per participant. The participant was requested to focus on a point inside the eyepiece and open their eyes as wide as possible. They were asked to avoid rapid and frequent blinking of eyes, movement of eyes, and specifically to avoid blinking during stimulus presentation. The videos were then post-processed to obtain the dilation of the pupil. The pupil was extracted as an ellipse from each frame with a segmentation algorithm and the diameter was calculated from the average of the major and minor axes of the resulting pupil ellipse. The difference between the maximum dilation and maximum constriction of both pupils in response to each color for every trial was calculated for each person immediately before and after NITESGON. The pupil size was measured as a direct response to the NITESGON and did not include a specific task.”

“Procedure: Participants performed the auditory oddball task twice, once immediately before and once immediately after the NITESGON session. In addition, saliva and pupil dilation were also collected immediately before and immediately after the NITESGON session. The reason why we collected auditory oddball task and pupil size before and after NITESGON is to avoid a direct effect of the learning taks or inference of the current sent to the scalp. Participants were randomly assigned to the active or sham NITESGON group. The researcher who controlled the NITESGON device was not involved in instructing the participant; this was performed by a second researcher who was blind to the stimulation protocol.”

2) I'd still recommend reporting separate correlations by group. Many of the correlation effects appear to uncouple in the stimulation condition, which could alter some interpretations of the data. If this happens, additional discussion is warranted.

We added the correlations for each group separately for each of the experiments.

3) The addition of the "before" and "after" (blue/yellow colors) in some of the keys was a useful change. However, these names could be more descriptive. It's not readily apparent what these mean when the other variable also describes timepoints (e.g., during stimulation).

We included this in the paper by indicating that before or after NITESGON.

4) There are quite a number of grammatical and spelling errors throughout, which I won't point out individually. The paper should be proofread again and revised before final acceptance.

I asked a native speaker to double check for typo.

5) It would be useful to foreground the "competition" effects of tag-and-capture in the Results section for the interference studies. This is described well in the discussion, but the same point should be repeated higher up. Otherwise, this buries the lead for the reader.

We include this in the paper under experiment 4. “Experiments 2 and 3 revealed both retroactive and proactive memory effects 7-days after initial learning of the two tasks. To further explore if NITESGON is linked to behavioral tagging and evaluate if interference impacts NITESGON as the strong stimulus, Experiment 4 removed the object-location task used in Experiments 2 and 3 and replaced it with a Japanese-English verbal associative learning task similar to the Swahili-English verbal associative task. Considering that memory formation and persistence are susceptible to interference occurring pre-and post-encoding(39-41) and are heavily influenced by commonality amongst the learned and intervening stimuli(42); it is believed that conducting two consecutive, like-minded word-association (i.e., Swahili-English and Japanese-English) tasks will result in one’s consolidation process interfering with that of the other(43). Furthermore, research on the synaptic tag-and-capture hypothesis suggest that memory interference is the result of synaptic competition, a proposed “fight for proteins” that arises between tagged synapses amongst limited proteins that leads to one memory converting to long-term memory at the expense of the other(68). Considering how our previous experiments suggest the effect obtained by NITESGON improves the consolidation of information via behavioral tagging, it is possible that NITESGON on the first task might help reduce the overall interference effect on the second task.”